# In vivo characterization of the bacterial intramembrane-cleaving protease RseP using the heme binding tag-based assay iCliPSpy

Thomas Kupke [1✉], Rabea M. Götz [2], Florian M. Richter[2], Rainer Beck [1], Fabio Lolicato [1,3], Walter Nickel [1], Carsten Hopf [2] & Britta Brügger [1✉]

Regulated intramembrane proteolysis (RIP) describes the protease-dependent cleavage of transmembrane proteins within the hydrophobic core of cellular membranes. Intramembrane-cleaving proteases (I-CliPs) that catalyze these reactions are found in all kingdoms of life and are involved in a wide range of cellular processes, including signaling and protein homeostasis. I-CLiPs are multispanning membrane proteins and represent challenging targets in structural and enzyme biology. Here we introduce iCLiPSpy, a simple assay to study I-CLiPs in vivo. To allow easy detection of enzyme activity, we developed a heme-binding reporter based on TNFα that changes color after I-CLiP-mediated proteolysis. Co-expression of the protease and reporter in *Escherichia coli* (*E. coli*) results in white or green colonies, depending on the activity of the protease. As a proof of concept, we use this assay to study the bacterial intramembrane-cleaving zinc metalloprotease RseP in vivo. iCLiPSpy expands the methodological repertoire for identifying residues important for substrate binding or activity of I-CLiPs and can in principle be adapted to a screening assay for the identification of inhibitors or activators of I-CLiPs, which is of great interest for proteases being explored as biomedical targets.

[1] Heidelberg University Biochemistry Center (BZH), Im Neuenheimer Feld 328, 69120 Heidelberg, Germany. [2] Center for Mass Spectrometry and Optical Spectroscopy (CeMOS), Mannheim University of Applied Sciences, Paul-Wittsack-Str. 10, 68163 Mannheim, Germany. [3] Department of Physics, University of Helsinki, Helsinki, Finland. ✉email: thomas.kupke@bzh.uni-heidelberg.de; britta.bruegger@bzh.uni-heidelberg.de

-CLiPs cleave transmembrane proteins within the lipid bilayer (reviewed in[1]). This type of proteolytic activity is conserved throughout all domains of life, and contributes to a wide range of cellular processes, including protein maturation and degradation, and cellular signaling. I-CLiPs can be divided into different groups depending on the type of catalysis. Although these proteases generally have low substrate specificity, some exclusively cleave type I transmembrane proteins, such as mammalian gamma-secretase[2], which is associated with Alzheimer's disease, while other proteases target type II transmembrane proteins. Examples of the latter are members of the SPPL2 (signal peptide peptidase-like 2) family, whose best-known substrate is tumor necrosis factor alpha (TNFα), a key regulator of the inflammatory response[3]. I-CLiPs are involved in highly coordinated RIP signaling events. In general, signal transduction by RIP begins with cleavage of the extracellular domain of transmembrane proteins in a process known as ectodomain shedding, and then the remaining membrane-bound fragment is processed by I-CLiPs[4–6]. A well-characterized I-CLiP in E. coli is the inner membrane zinc metalloprotease RseP[7–9] (formerly called YaeL or EcfE). RseP activates the cell envelope stress response in E. coli by cleaving the anti-$\sigma^E$ type II transmembrane protein RseA. This signaling cascade is initiated by the membrane-bound serine protease DegS[10–12], which acts as a sheddase and removes the periplasmic domain of RseA. The remaining N-terminal fragment 1-148 of RseA is then cleaved by RseP between residues 108 and 109[13].

RseP is a transmembrane protein with four transmembrane helices and two tandemly arranged periplasmic PDZ domains that serve as size exclusion filters for the substrate[8,9,14–17]. The $Zn^{2+}$ ion required for activity is bound by the side chains of His22, His26 and Asp402[7,8,18]. An important structural element involved in substrate recognition and cleavage is the membrane-embedded β-hairpin-like structure formed by residues Ile61 to Glu75 [membrane-reentrant β-loop, β-MRE][19]. In addition, the GFG motif of RseP and a membrane-bound amphiphilic helix (residues Pro323 to Thr350) contribute to substrate discrimination and binding[20,21]. The recently deposited crystal structure of RseP with the inhibitor batimastat provides mechanistic insights into intramembrane proteolysis and suggests a gating mechanism to regulate substrate entry[22].

RseP also cleaves signal peptides generated by leader peptidase-mediated processing of presecretory proteins[9,23,24]. This low substrate specificity prompted us to investigate whether reporter substrates could be used to study RseP in vitro and in vivo. Ideally, such a reporter would change its spectroscopic characteristics upon proteolytic processing and also contain a tag that would allow affinity purification and detection.

In a recent study, we used an E. coli expression model to identify TNFα, its proteolytical processing products and other substrates of SPPL2a/b proteases as heme (= heme B) binding proteins[25]. In vivo, TNFα is cleaved by the sheddase TACE, releasing an N-terminal fragment (NTF) and a soluble cytokine[26]. The NTF is further processed by the intramembrane-cleaving proteases SPPL2a/b, resulting in two intracellular domains (ICDs) that differ in length [TNFα-(1-39) and TNFα-(1-34)[27]]. We have described that heme binding depends on both a cytosolic cysteine residue proximal to the membrane (Cys30) and oligomerization of the transmembrane domain (TMD). Proteolytic processing of the N-terminal fragment of TNFα alters the nature of ferric heme binding, switching from stable, hexa-coordinated ligation by two Cys30 side chains in dimeric TNFα-(1-39) (so-called bis-thiolate ligation) to low affinity, penta-coordinated heme bound by only one Cys30 side chain in the shorter, monomeric ICD TNFα-(1-34)[25] (Fig. 1a). Deletion of five of the nine TMD residues still

present in TNFα-(1-39) (residues Leu35 to Leu39) removes the dimerization motif $S^{34}XXS^{37}$ and thus prevents heme promoted dimerization of TNFα-(1-34) (Fig. 1a). Bis-thiolate ligated ferric heme shows a split Soret band with absorbance maxima at about 370 nm and 450 nm and an additional absorbance maximum at about 550 nm, so that TNFα-(1-39) with bound heme is a green colored peptide. In contrast, TNFα-(1-34) has only one Soret band at about 370 nm and appears uncolored due to its low heme binding affinity[25] (Fig. 1a).

In the present study, we exploit the different spectroscopic properties of substrate TNFα-(1-39) and product TNFα-(1-34) to investigate intramembrane processing catalyzed by RseP in vivo. We show that E. coli cells co-producing an MBP-TNFα-(1-39) fusion protein (MBP, maltose binding protein) optimized for iCliPSpy and an active RseP protease are uncolored/white, while cells expressing the inactive RseP enzyme together with the reporter are green. To confirm the processing of TNFα-(1-39) catalyzed by RseP, we used mass spectrometry. By determining the intact mass and additionally the sequence of tryptic fragments of the processed substrate proteins, we show that RseP trims tripeptides and cleaves L/F-S peptide bonds within the substrate used. To test whether this assay for I-CLiPs allows the identification of residues within RseP that are important for activity and/or substrate binding, we generated mutant rseP genes by error-prone PCR[28]. Using this approach, we identified Tyr69 as a critical residue for RseP activity and/or substrate binding. Molecular dynamics (MD) simulations of the AlphaFold[29,30] structure of RseP (with bound $Zn^{2+}$ ion) in a lipid bilayer with a lipid composition mimicking E. coli membranes show that residue Tyr69 is located in the membrane/cytosol environment near the active site and may play a role in positioning the substrate peptide through hydrophobic interactions. These results are in agreement with the recently published crystal structure of RseP and the model proposed for substrate binding, in which the so-called edge strand of RseP (residues Gly67 to Val70 including Tyr69 and part of the β-MRE) forms an antiparallel β-sheet with the substrate TMD[22]. Our data show that iCliPSpy is a simple and sensitive assay to identify and study I-CLiP activities in vivo.

## Results

**Substrate and product of the reaction catalyzed by RseP.** An effective production strategy for the heme-binding peptide TNFα-(1-39) as potential substrate of RseP and other I-CLiPs, is its expression as MBP fusion protein in E. coli[25]. To ensure the correct topology [considering that full-length TNFα is a type II transmembrane protein: N-terminus in (cytoplasm) and C-terminus out (periplasm/lumen of the ER)], MBP was fused to the N-terminus of TNFα-(1-39) without the signal peptide (residues -24 to -1, Fig. 1b). TNFα-(1-39) contains only the N-terminal nine amino acid residues $L^{31}$FLSLFSFL$^{39}$ of the TNFα TMD and is therefore found in both the membrane and cytosolic fractions[25,27]. Instead of the MBP construct described in our recent study[25], we used a mutant MBP protein (MBP$_{mut}$) containing additional substitutions, including I317V, which further increases the affinity for amylose[31] (Fig. 1b). To increase the amount of heme bound by TNFα, we used the variant peptide TNFα-(1-39)-L31P, which shows a stronger interaction with heme compared to the wildtype peptide[25]. In addition, MBP$_{mut}$ and TNFα-(1-39)-L31P are linked in MBP$_{mut}$-TNFα-(1-39)-L31P by a shortened linker peptide (Fig. 1b), which should increase the proteolytic stability of the fusion protein. We expressed the putative product of RseP-mediated intramembrane proteolysis, TNFα-(1-34), in the same manner [MBP$_{mut}$-TNFα-(1-34)-L31P]. The *MBP$_{mut}$-TNFα-(1-39)-L31P, MBP$_{mut}$-TNFα-(1-34)-L31P* and *His-MBP-TNFα-(1-39)-L31P* genes (Fig. 1b) were cloned into the

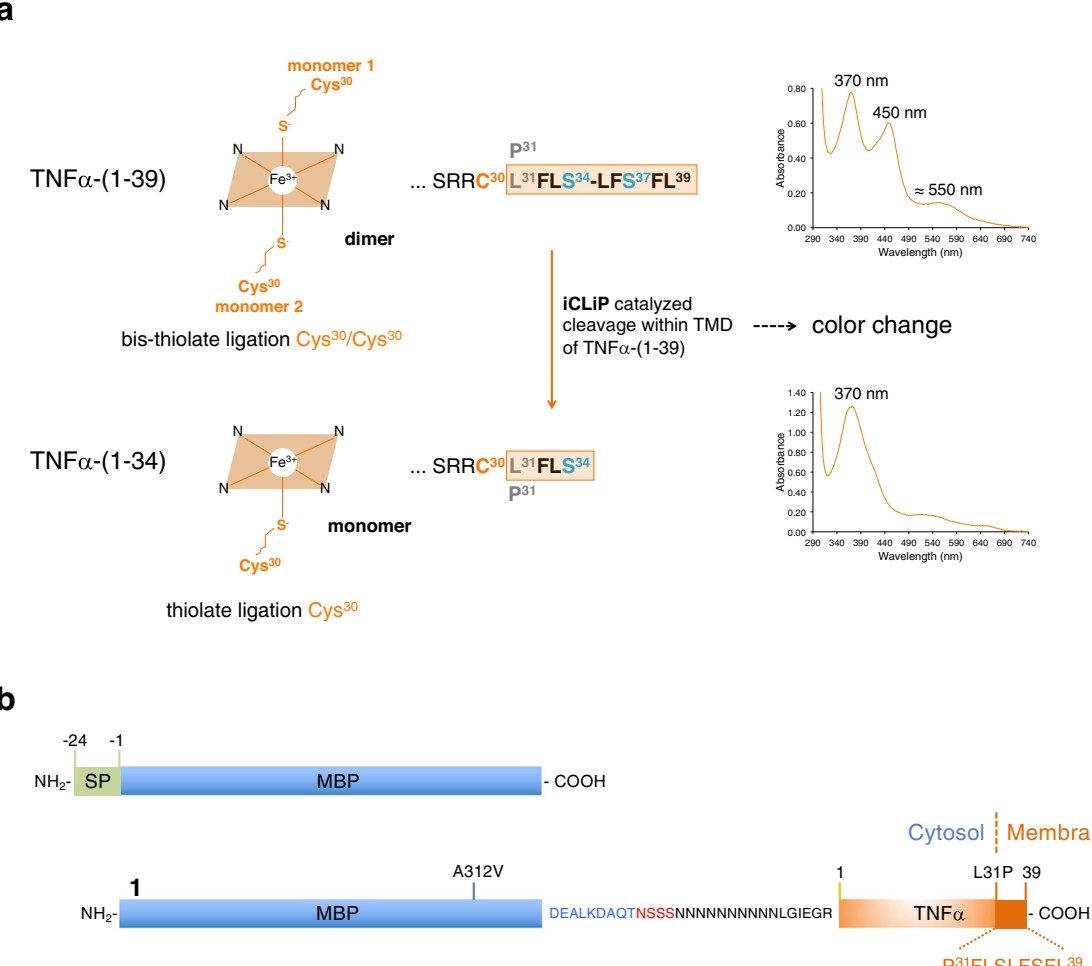

**Fig. 1 The principle of the heme binding tag-based iCliPSpy assay. a** Bis-thiolate ligation of heme depends on dimerization of TNFα-(1-39). Cleavage within the TMD part L31-L39 of TNFα-(1-39) prevents dimerization by removing or disrupting the $S^{34}XXS^{37}$ motif and results in mono-thiolate ligation of heme by Cys30. Bis-thiolate and mono-thiolate-coordinated heme molecules differ in their absorbance spectra, so that TNFα-(1-39) (substrate for intramembrane-cleaving protease) and TNFα-(1-34) (product of intramembrane proteolysis) with bound heme show different colors. Because of increased heme binding affinity, we use TNFα-(1-39)-L31P instead of TNFα-(1-39) wt as heme binding tag. **b** MBP-TNFα constructs used in our studies [lacking the signal peptide (SP) of *E. coli* MBP shown at the top of the figure]. **1**, MBP- TNFα-(1-39)-L31P with a poly-Asn linker was used in our initial study to characterize heme binding of TNFα[25]. **2-4** MBP-TNFα fusions used in the present study all contain the peptide DAALAAAQTNAAA, which links MBP to TNFα-(1-39). The topology of MBP-TNFα-(1-39) fusion proteins is exemplary shown for construct 1. For co-production with I-CLiP RseP, construct **3** [named MBP_mut-TNFα-(1-39)-L31P] was used, which contains additional eight substitutions that further increase affinity for amylose, reduce the surface entropy and increase the crystal-packing interactions between adjacent MBP molecules[31].

pETDuet-1 vector (Supplementary Table 1 and Supplementary Note 1) and the recombinant plasmids were transformed into *E. coli* T7 Express cells. For in vivo experiments, expression of MBP fusions was performed without IPTG induction on agar plates (Fig. 2a). A general disadvantage of the T7 expression system is the already existing low protein expression without IPTG induction, also called leaky expression. However, leaky expression can be an advantage, especially in the expression of

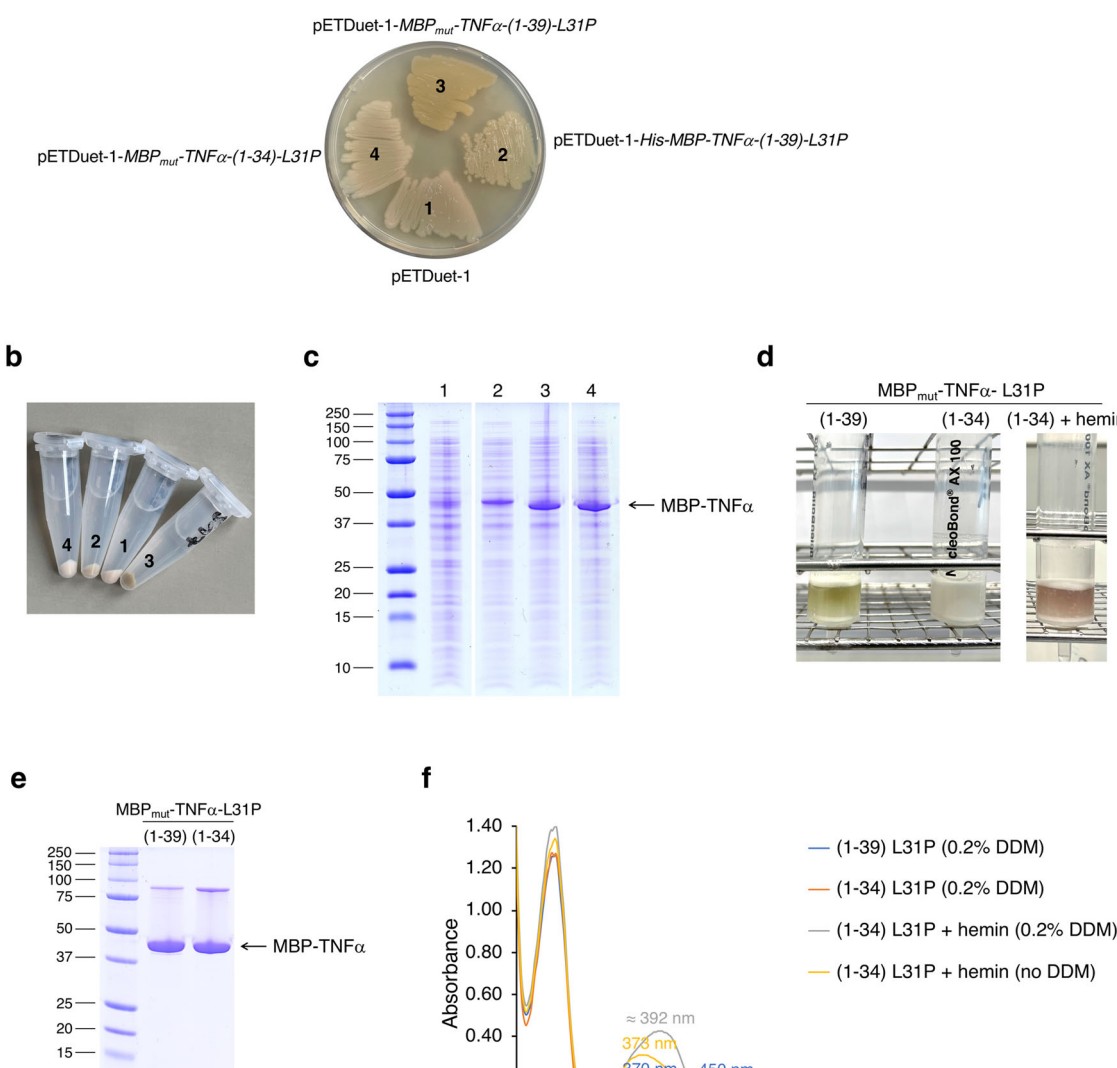

**Fig. 2 Characterization of substrate and product of the iCliPSpy assay. a** *E. coli* T7 Express cells harboring the plasmids **1**, pETDuet-1 (control), **2**, pETDuet-1-*His-MBP-TNFα-(1-39)-L31P* (expressing a substrate for I-CLiPs), **3**, pETDuet-1-*MBP_mut-TNFα-(1-39)-L31P* (expressing an alternative substrate for I-CLiPs) or **4**, pETDuet-1-*MBP_mut-TNFα-(1-34)-L31P* (expressing the putative product of intramembrane proteolysis), were incubated overnight on an agar plate at 37 °C followed by another day at room temperature. **b** *E. coli* T7 Express cells harboring the plasmids **1**, pETDuet-1, **2**, pETDuet-1-*His-MBP-TNFα-(1-39)-L31P*, **3**, pETDuet-1-*MBP_mut-TNFα-(1-39)-L31P* or **4**, pETDuet-1-*MBP_mut-TNFα-(1-34)-L31P*, were grown on an agar plate for six days at room temperature, scraped from the plate, resuspended in TN buffer, and harvested by centrifugation and **c** then further analyzed by SDS-PAGE (after solubilization by boiling in Laemmli sample buffer). **d** MBP_mut-TNFα-(1-39)-L31P and MBP_mut-TNFα-(1-34)-L31P were purified by amylose affinity chromatography from the membrane fraction using DDM as solubilizing detergent (left and middle columns) from IPTG induced *E. coli* T7 Express cells harboring the above-listed plasmids. Purification of MBP_mut-TNFα-(1-34)-L31P was repeated after addition of 25 μM hemin (right column) to membrane proteins solubilized with DDM. Purified MBP fusion proteins were then analyzed by **e** SDS PAGE and **f** UV/Vis spectroscopy [plus 0.2% DDM: blue trace, MBP_mut-TNFα-(1-39)-L31P; red trace, MBP_mut-TNFα-(1-34)-L31P; grey trace, MBP_mut-TNFα-(1-34)-L31P plus hemin; minus DDM: yellow trace, MBP_mut-TNFα-(1-34)-L31P plus hemin]. When DDM is removed from hemin reconstituted MBP_mut-TNFα-(1-34)-L31P by washing the amylose column with buffer containing no DDM, the typical absorbance maximum of about 373 nm for mono-thiolate ligated heme is observed[25,56], showing that presence of DDM leads to a bathochromic shift of the absorbance maximum from 373 nm to about 392 nm for mono-thiolate ligated heme. Free hemin in the elution buffer used with 0.2% DDM has an absorbance maximum at 405 nm. In vivo detection of MBP-TNFα fusion proteins, as shown in **a**, was repeated once (and in a different way in Fig. 3 a and Supplementary Fig. 2a), and one representative result is shown. Detection of MBP-TNFα fusion proteins in cells by SDS-PAGE (**b** and **c**) was done once, and the UV/Vis spectra of purified MBP_mut-TNFα-(1-39)-L31P and MBP_mut-TNFα-(1-34)-L31P fusion proteins (**f**), were checked in a second experiment.

membrane proteins, as the low-level expression prevents the formation of inclusion bodies[32,33] and also favors heme-binding of TNFα-(1-39)-L31P, as the heme concentration in cells is extremely low[34].

After several days at room temperature cells expressing MBP_mut-TNFα-(1-39)-L31P were ocher green, while cells expressing His-MBP-TNFα-(1-39)-L31P were only faintly colored and cells expressing MBP_mut-TNFα-(1-34)-L31P or the empty vector control

pETDuet-1 were white (Fig. 2a, b). SDS-PAGE analysis of whole cells scraped from agar plates confirmed the production of the different MBP-TNFα fusion proteins and showed that reduced expression of His-MBP-TNFα-(1-39)-L31P is responsible for the less intense color of these cells (Fig. 2c). Therefore, we selected the MBP$_{mut}$-TNFα-(1-39)-L31P, and MBP$_{mut}$-TNFα-(1-34)-L31P constructs for further experiments. The membrane-associated MBP fusion proteins were then solubilized with the non-ionic detergent n-dodecyl-β-D-maltopyranoside (DDM) and purified by amylose affinity chromatography in the presence of DDM. UV/Vis spectroscopy of the purified fusion proteins confirmed the ligation of heme by bis-thiolate in MBP$_{mut}$-TNFα-(1-39)-L31P and by thiolate in MBP$_{mut}$-TNFα-(1-34)-L31P (Fig. 2d-f). MBP$_{mut}$-TNFα-(1-34)-L31P with thiolate ligated heme shows an absorbance maximum at about 392 nm, as the addition of DDM results in a bathochromic shift of the absorbance maximum by ∼20 nm (Fig. 2f). Cells expressing MBP$_{mut}$-TNFα-(1-34)-L31P are white because the heme content is very low. Heme reconstitution of MBP$_{mut}$-TNFα-(1-34)-L31P, however, resulted in a red-brown color of the fusion protein (Fig. 2d, f). Taken together, our results show that recombinantly expressed reporter substrate and mimicked product of I-CliP-mediated proteolysis in E. coli differ in their spectroscopic properties, resulting in visible changes in the coloring of bacterial colonies. Next, we tested whether our reporter MBP$_{mut}$-TNFα-(1-39)-L31P also serves as a substrate for bacterial I-CLiPs.

**Visualization of the in vivo activity of RseP.** A well-characterized bacterial I-CLiP is E. coli RseP, which we used to develop the I-CLiP assay iCliPSpy. A previously published in vivo assay for monitoring RseP activity[9,23] uses E. coli rseP rseA double-disrupted cells carrying plasmids expressing RseP-Myc[8] and an HA-MBP-tagged, truncated version of RseA, RseA140, as a substrate for RseP. RseA140 cleavage by RseP was monitored by immunoblot analysis[9,23] and this assay has been successfully used to study the structure, function and substrate binding of RseP by site-directed mutagenesis[19-21,35]. Since the in vivo activity of RseP is not directly visible when looking at colonies of E. coli cells over-expressing RseP-Myc and HA-MBP-RseA140, we sought to develop an assay that allows simple detection of I-CLiP activity based on the color of the colonies.

To test whether MBP$_{mut}$-TNFα-(1-39)-L31P can be cleaved by RseP, RseP-Myc[8] wild type (wt, Supplementary Fig. 1a) or the catalytically inactive variant RseP-Myc H22F were expressed together with our I-CLiP reporter. Since RseP is essential for bacterial growth[7], we performed iCliPSpy in presence of the chromosomal copy of rseP. In principle, the RseP-dependence of growth can be circumvented by down-regulating the porins OmpA and OmpC[36]. Our data, however, show that the amount of chromosomally encoded RseP is not sufficient to degrade overproduced MBP$_{mut}$-TNFα-(1-39)-L31P to any appreciable extent (Fig. 3a), so that additional genetic modification was not required.

Co-expression of the I-CliP reporter with RseP-Myc wt resulted in white cells (Fig. 3a), indicating that the overexpressed RseP had cleaved TNFα-(1-39)-L31P. The H22F substitution in RseP, which leads to a loss of zinc binding, inhibited RseP activity, so that co-expression of MBP$_{mut}$-TNFα-(1-39)-L31P with rseP-Myc H22F resulted in green E. coli colonies (Fig. 3a). SDS-PAGE analysis of whole cells scraped from the agar plate confirmed comparable expression levels of the MBP$_{mut}$-TNFα fusions and showed that the RseP cleavage product of MBP$_{mut}$-TNFα-(1-39)-L31P has a similar apparent molecular weight as MBP$_{mut}$-TNFα-(1-34)-L31P, indicating that only a few amino acid residues were cleaved off by RseP (Fig. 3b). Expression of

rseP-Myc wt and rseP-Myc H22F genes was confirmed by western blotting (Fig. 3c). Higher expression levels of rseP-Myc H22F compared to rseP-Myc wt were less pronounced in case of co-expression with MBP$_{mut}$-TNFα-(1-39)-L31P (compare the first two lanes with lanes five and six of the immunoblot shown in Fig. 3c). As expected and despite this higher expression, proteolytic cleavage of MBP$_{mut}$-TNFα-(1-39)-L31P by RseP-Myc H22F was not observed. In vivo processing of MBP$_{mut}$-TNFα-(1-39)-L31P was also confirmed for untagged RseP (Supplementary Figs. 1a-c and 2a, b). However, when a 10x HisTag (His$_{10}$) was fused directly without a linker peptide to the carboxy terminus of RseP, the activity of RseP was decreased (Supplementary Fig. 1a-c), suggesting that the interaction of the periplasmic His$_{10}$ Tag with the PDZ domains (Supplementary Fig. 1b) could affect folding, stability or activity of RseP.

In general, our results suggest that green-colored colonies appear when sufficient MBP$_{mut}$-TNFα-(1-39)-L31P is present. White colonies, however, are observed when MBP$_{mut}$-TNFα-(1-39)-L31P has been cleaved or when the reporter is expressed at very low concentrations. Therefore, an expression rate of the I-CLiP reporter that is independent of the sequence of the co-expressed rseP gene (Fig. 3b and Supplementary Fig. 2b) is an essential requirement for this assay.

**Processing changes heme binding properties of the substrate.** Our data suggest that intramembrane proteolysis catalyzed by RseP affects the dimerization motif S$^{34}$LFS$^{37}$ of TNFα-(1-39) and consequently alters the heme binding mode of the TNFα peptide, consistent with the color change. To test whether MBP$_{mut}$-TNFα-(1-39)-L31P indeed serves as a substrate for RseP cleavage, membrane-associated proteins solubilized with DDM from IPTG-induced E. coli cells co-expressing MBP$_{mut}$-TNFα-(1-39)-L31P with either rseP wt or rseP H22F were enriched via amylose affinity chromatography (Fig. 4a), analyzed by SDS-PAGE (Fig. 4b) and the heme coordination mode was determined by UV/Vis spectroscopy (Fig. 4c). The ratio of active and correctly folded protease to substrate may differ after IPTG-induced overexpression compared to leaky expression in the in vivo assay. This would lead to different proportions of fully processed substrate. Indeed, we observed that both amylose affinity columns were green, but the color was more intense in the case of co-expression with rseP H22F (Fig. 4a). MBP$_{mut}$-TNFα-L31P proteins purified from rseP wt over-expressing cells showed a slightly lower apparent molecular weight than MBP$_{mut}$-TNFα-L31P proteins purified from rseP H22F over-expressing cells (Fig. 4b). The UV/Vis spectrum of MBP$_{mut}$-TNFα-L31P purified from rseP H22F over-expressing cells showed absorbance maxima at 371 nm and 451 nm, demonstrating bis-thiolate ligation and confirming that RseP H22F does not cleave MBP$_{mut}$-TNFα-(1-39)-L31P. For MBP$_{mut}$-TNFα-L31P purified from rseP wt over-expressing cells, however, the absorbance at 451 nm was significantly reduced, the absorbance maximum at 371 nm was shifted to 373 nm and the absorbance at 392 nm was increased [Fig. 4c, see also Fig. 2f, which shows a UV/Vis absorbance spectrum of the putative RseP cleavage product MBP$_{mut}$-TNFα-(1-34)-L31P, which would be expected if proteolysis was complete and the dimerization motif disrupted]. As expected, unprocessed and RseP-processed MBP$_{mut}$-TNFα-(1-39)-L31P fusion proteins were also present in the cytosolic fraction (Supplementary Fig. 3a-e; compare also Kupke et al.[25]).

Solubilized membrane-associated MBP$_{mut}$-TNFα-L31P proteins purified from E. coli co-overexpressing either inactive rseP H22F or active rseP wt were then separated by gel filtration in presence of DDM into heme-binding (monomeric or dimeric) MBP$_{mut}$-TNFα-L31P and heme-free monomeric MBP$_{mut}$-TNFα-

**a**

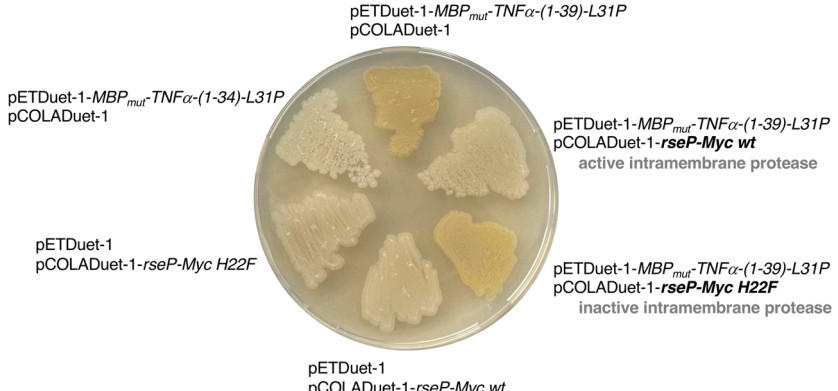

**b**

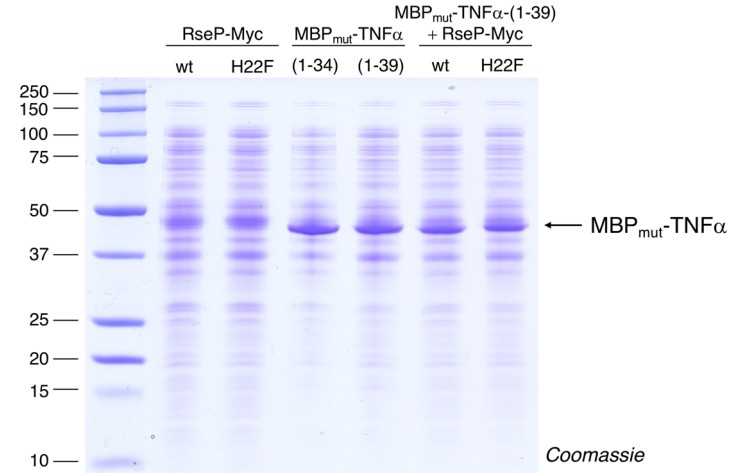

**c**

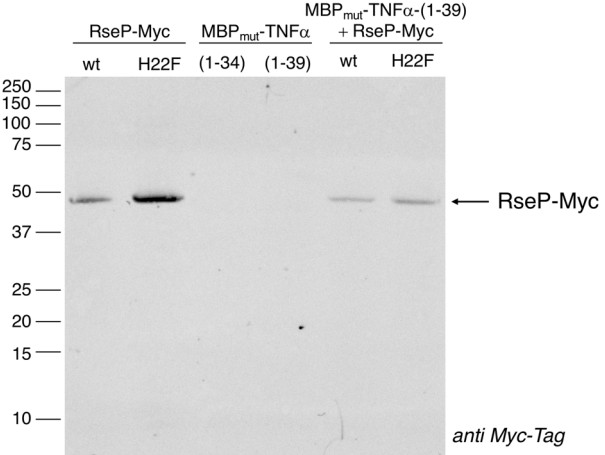

L31P fusion proteins (Fig. 4d). Heme-free monomers are always present as heme is not stoichiometrically bound by TNFα-ICDs[25]. The higher content of heme-free monomers of MBP$_{mut}$-TNFα-L31P in the case of RseP wt compared to RseP H22F is probably related to a lower heme binding affinity of the processed fusion protein (compare UV/Vis spectra in Fig. 4c), because hydrophobic residues within the heme-binding motif SRRCPFLSLFSFL of TNFα may also contribute to heme binding and because bis-thiolate ligation stabilizes heme binding[25]. The heme-binding MBP$_{mut}$-TNFα-L31P fusion protein obtained in the case of RseP H22F eluted at 15.40 ml, whereas the protein observed for RseP wt eluted at 16.05 ml (corresponding to a lower apparent molecular weight) from the size exclusion column (Fig. 4d). Since MBP$_{mut}$-TNFα-(1-39)-L31P contains only a single cysteine

**Fig. 3 Visualization of in vivo intramembrane proteolysis catalyzed by RseP-Myc. a** *E. coli* T7 Express cells harboring the two compatible plasmids pETDuet-1 and pCOLADuet-1, which co-express genes encoding the heme-binding protein MBP$_{mut}$-TNFα-(1-39)-L31P and the active RseP-Myc wt or inactive RseP-Myc H22F protease, were incubated for five days at room temperature on an agar plate. Control cells expressed only one gene encoding either RseP-Myc wt, RseP-Myc H22F, the substrate MBP$_{mut}$-TNFα-(1-39)-L31P or the putative proteolysis product MBP$_{mut}$-TNFα-(1-34)-L31P. Green *E. coli* colonies are only observed if the substrate present, MBP$_{mut}$-TNFα-(1-39)-L31P, is not C-terminally truncated by RseP. **b** and **c** Analysis of *E. coli* T7 Express cells expressing genes for RseP-Myc wt, RseP-Myc H22F, MBP$_{mut}$-TNFα-(1-34)-L31P, MBP$_{mut}$-TNFα-(1-39)-L31P or coexpressing genes for MBP$_{mut}$-TNFα-(1-39)-L31P and RseP-Myc wt/H22F, respectively, and grown on a selection plate for four days at room temperature by **b**, SDS-PAGE, and **c**, by immunoblotting using a monoclonal anti-Myc-Tag antibody. Comparative in vivo analysis of RseP activities as shown in **a** and **b** was repeated with untagged RseP proteases with the same result (compare Supplementary Fig. 2).

residue [Cys30 of TNFα-(1-39)-L31P,] bis-thiolate ligation of heme requires dimerization[25] (Fig. 1a). Therefore, the eluted proteins were again characterized by UV/Vis spectroscopy to distinguish between monomeric and dimeric heme-binding proteins (Fig. 4e). The UV/Vis spectra of the heme-binding proteins purified by gel filtration confirm that dimeric MBP$_{mut}$-TNFα-(1-39)-L31P is present when *MBP$_{mut}$-TNFα-(1-39)-L31P* was co-expressed with *rseP H22F* but not with *rseP wt*. In the latter case, the absorbance maximum at about 392 nm (measured in the presence of DDM) shows that processed MBP$_{mut}$-TNFα-L31P coordinates heme *via* a single cysteine residue and is monomeric. It appears that RseP-processed (monomeric) heme binding MBP$_{mut}$-TNFα-L31P is not uniform, as the observed elution volume is as mentioned either 16.05 ml (when measuring the absorbance at 280 nm) or 15.85 ml (absorbance at 371 nm; Fig. 4d). We assume that at least two different processing products with different heme binding affinities (and thus different heme content) are present. At higher protein concentrations, a processing product that still contains part of the dimerization motif may dimerize to a small extent and therefore also shows an absorbance maximum at about 450 nm (and 373 nm; compare spectrum "+ RseP wt" of Fig. 4e with the corresponding spectrum of Fig. 4c).

In summary, gel filtration shows that cleavage by RseP results in MBP$_{mut}$-TNFα fusion proteins that differ in molecular weight and heme binding properties from the substrate MBP$_{mut}$-TNFα-(1-39)-L31P.

**RseP processing sites in the MBP$_{mut}$-TNFα substrate.** The UV/Vis absorbance spectrum of the MBP$_{mut}$-TNFα-(1-39)-L31P fusion protein purified from cells co-overexpressing *rseP wt* shows that the heme-coordinating residue Cys30 is still present in the proteolytic cleavage product, but that the bis-thiolate ligation of heme has been largely lost. Consequently, cleavage occurred within TNFα TMD residues 31-39 of the reporter protein, disrupting the dimerization motif (compare Fig. 1). To determine the exact RseP processing site(s) within the C-terminal TNFα TMD residues P$^{31}$FLSLFSFL$^{39}$ by mass spectrometry, *MBP$_{mut}$-TNFα-(1-39)-L31P* was again co-expressed with either *rseP wt*, empty vector control pCOLADuet-1, *rseP-Myc wt* or *rseP-Myc H22F*. The MBP-TNFα fusion proteins produced were purified from the membrane fraction by amylose affinity chromatography and then analyzed by SDS-PAGE (Fig. 5a, b) and UV/Vis spectroscopy (Fig. 5c) to confirm the experimental data shown in Fig. 4 and to ensure RseP catalyzed processing within TNFα-(1-39)-L31P prior to mass spectrometric analysis. As previously observed, the MBP$_{mut}$-TNFα fusion proteins that still bind heme *via* bis-thiolate ligation have a slightly higher molecular weight than the fusion proteins that bind heme *via* mono-thiolate ligation (Fig. 5a, b), but the resolution of SDS-PAGE is too low to separate differently truncated proteins. Two complementary mass spectrometric analyses were then performed to determine the proteolytic cleavage sites: (i) analysis of the intact mass of the MBP-TNFα fusion proteins (Fig. 5d, e and Supplementary Fig. 4a-d). and (ii) sequence analysis and quantification of their tryptic peptides (Supplementary Fig. 4e). For the first approach, MBP-TNFα fusion proteins obtained from the four different genetic backgrounds were separated by reversed-phase (RP)-liquid chromatography (LC) and the intact mass information was obtained by coupled electrospray ionization mass spectrometry (ESI-MS) and compared with the calculated average molecular weights of C-terminally truncated MBP$_{mut}$-TNFα-(1-39)-L31P fusion proteins (Fig. 5d, e). The detected major degradation product MBP$_{mut}$- … CP$^{31}$ [lacking the terminal eight amino acid residues of TNFα-ICD-(1-39)] is not an RseP processing product, as it is also present in the absence of over-expressed active protease (Fig. 5e). Residues F$^{32}$LSLFSFL$^{39}$ could be cleaved off by unspecific proteolytic processing (independent of RseP), as to our knowledge there is no *E. coli* K12 endopeptidase that specifically cleaves Pro-X peptide bonds. We assume that MBP$_{mut}$- … CP$^{31}$ is detectable, despite the absence of TNFα TMD residues, because the membrane fraction has not been washed and therefore also contains soluble proteins (but also the hydrophobic heme cofactor bound to the CP motif might target this fusion protein to the membrane).

The full-length MBP$_{mut}$-TNFα-(1-39)-L31P fusion protein characterized by the C-terminal sequence … C$^{30}$PFLSLFSFL$^{39}$, however, was identified both by its intact mass and by sequence analysis of tryptic peptides (see Supplementary Note 2), for "Empty vector control" and "RseP-Myc H22F" (inactive RseP protein) purifications (Fig. 5e, and Supplementary Fig. 4a, b, e). This full-length MBP$_{mut}$-TNFα-(1-39)-L31P protein is able to bind heme by bis-thiolate ligation, resulting in the characteristic absorbance maxima of 370 nm, 450 nm and 550 nm[25,37,38].

However, when *MBP$_{mut}$-TNFα-(1-39)-L31P* was co-expressed with *rseP wt* or *rseP-Myc wt*, MBP$_{mut}$-TNFα-(1-39)-L31P ( = MBP$_{mut}$- … CPFLSLFSFL$^{39}$) was cleaved between residues F36 and S37 as well as between L33 and S34 of TNFα-(1-39)-L31P in both cases, resulting in the major processing products MBP$_{mut}$- … CPFLSLF$^{36}$ and MBP$_{mut}$- … CPFL$^{33}$ for both wt proteases (Fig. 5e, and Supplementary Fig. 4c-e). Upon co-expression with *rseP wt*, we found small amounts of three additional cleavage products, MBP$_{mut}$- … CPF$^{32}$, MBP$_{mut}$- … CPFLS$^{34}$ (Fig. 5e, and Supplementary Fig. 4c) and MBP$_{mut}$- … CPFLSLFS$^{37}$ (Supplementary Table 2). We do not know whether these truncated MBP$_{mut}$-TNFα-(1-39)-L31P proteins are also RseP processing products or whether the two major RseP processing products are further degraded by other *E. coli* proteases. MBP$_{mut}$- … CPFL$^{33}$ can no longer bind heme by bis-thiolate ligation, since the loss of the C-terminal pentapeptide L$^{35}$FSFL$^{39}$ already leads to mono-thiolate ligated heme[25] (Fig. 2). MBP$_{mut}$- … CPFLSLF$^{36}$, which lacks the second serine residue Ser37 important for (heme promoted) dimerization[25], could still dimerize to a small extent and bind heme by bis-thiolate ligation (compare absorbance of the processing products of RseP at 450 nm, Fig. 4c).

Small amounts of the major processing product of RseP, MBP$_{mut}$- … SRRCPFLSLF$^{36}$, were also identified for the "Empty

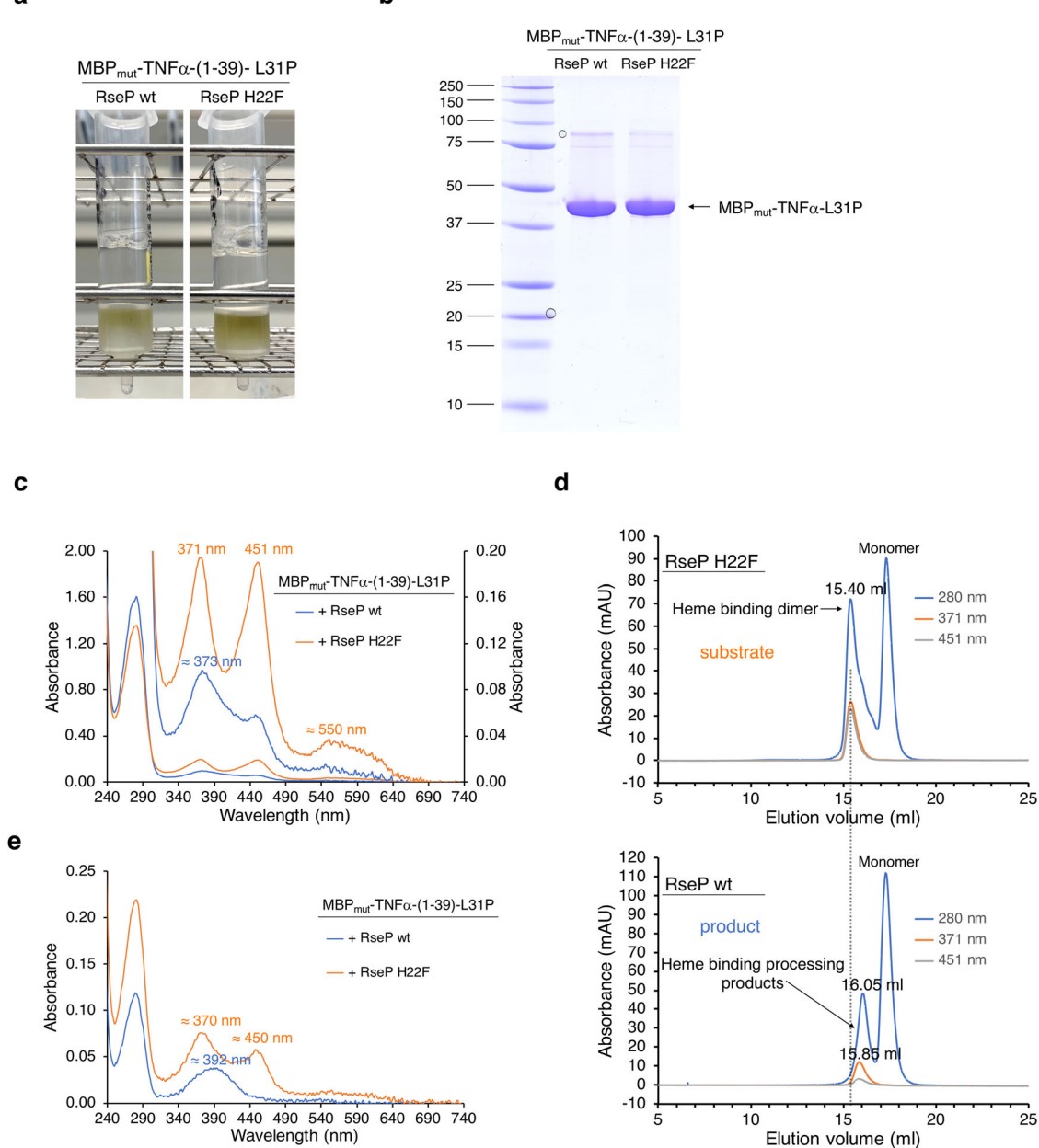

**Fig. 4 Characterization of the RseP cleavage product in vitro. a** Purification of membrane-associated MBP$_{mut}$-TNFα fusion proteins by amylose affinity chromatography from IPTG induced T7 cells co-expressing either genes for MBP$_{mut}$-TNFα-(1-39)-L31P and RseP wt (left column) or genes for MBP$_{mut}$-TNFα-(1-39)-L31P and mutant RseP H22F (right column). The purified MBP$_{mut}$-TNFα fusion proteins (which correspond to the product and substrate of the RseP reaction, respectively) were then analyzed **b** by SDS-PAGE, **c** by UV/Vis spectroscopy [blue trace, + RseP wt; orange trace, + RseP H22F; showing the full spectra (left absorbance axis) and the same spectra magnified (right absorbance axis)] and **d** by gel filtration on a Superose 6 Increase column [mAU = milli absorbance units; absorbance is monitored at 280 nm (blue trace), 371 nm (orange trace), and 451 nm (grey trace)]. As expected, the heme binding dimers and processing products (indicated by arrows) show different A$_{371nm}$/A$_{451nm}$ absorbance ratios. It can be assumed that heme binding to monomeric TNFα processing products significantly alters their shape/structure and thus their elution volume and apparent molecular weight. **e** UV/Vis spectroscopy of gel filtration fractions containing heme binding MBP$_{mut}$-TNFα fusion proteins (blue trace, + RseP wt, elution volume 15.4 ml – 16.2 ml; orange trace, + RseP H22F, elution volume 15.4 ml – 15.8 ml of the gel filtration runs shown in Fig. d). Purification of MBP$_{mut}$-TNFα proteins coproduced with either RseP wt or with RseP H22F (**a-c**) was performed four times and representative results are shown. **b-e** belong to the same experiment. Gel filtration (**d** and **e**) of the purified MBP$_{mut}$ fusion proteins was performed once.

vector control" but not for "RseP-Myc H22F" by LC-MS/MS analysis (Supplementary Table 2). It is likely that minor cleavage of the substrate protein by chromosomally encoded RseP (and other *E. coli* proteases) occurs, which can be detected by MS/MS after tryptic digest of purified MBP fusion proteins. This processing of MBP$_{mut}$-TNFα-(1-39)-L31P was not detected in

intact mass analysis (shown in Fig. 5e and Supplementary Fig. 4a).

In summary, mass spectrometric analysis confirms that RseP cleaves the TNFα substrate within the TMD residues 31-39. The different heme binding modes of substrate and processing product(s) determined by UV/Vis spectroscopy are demonstrably

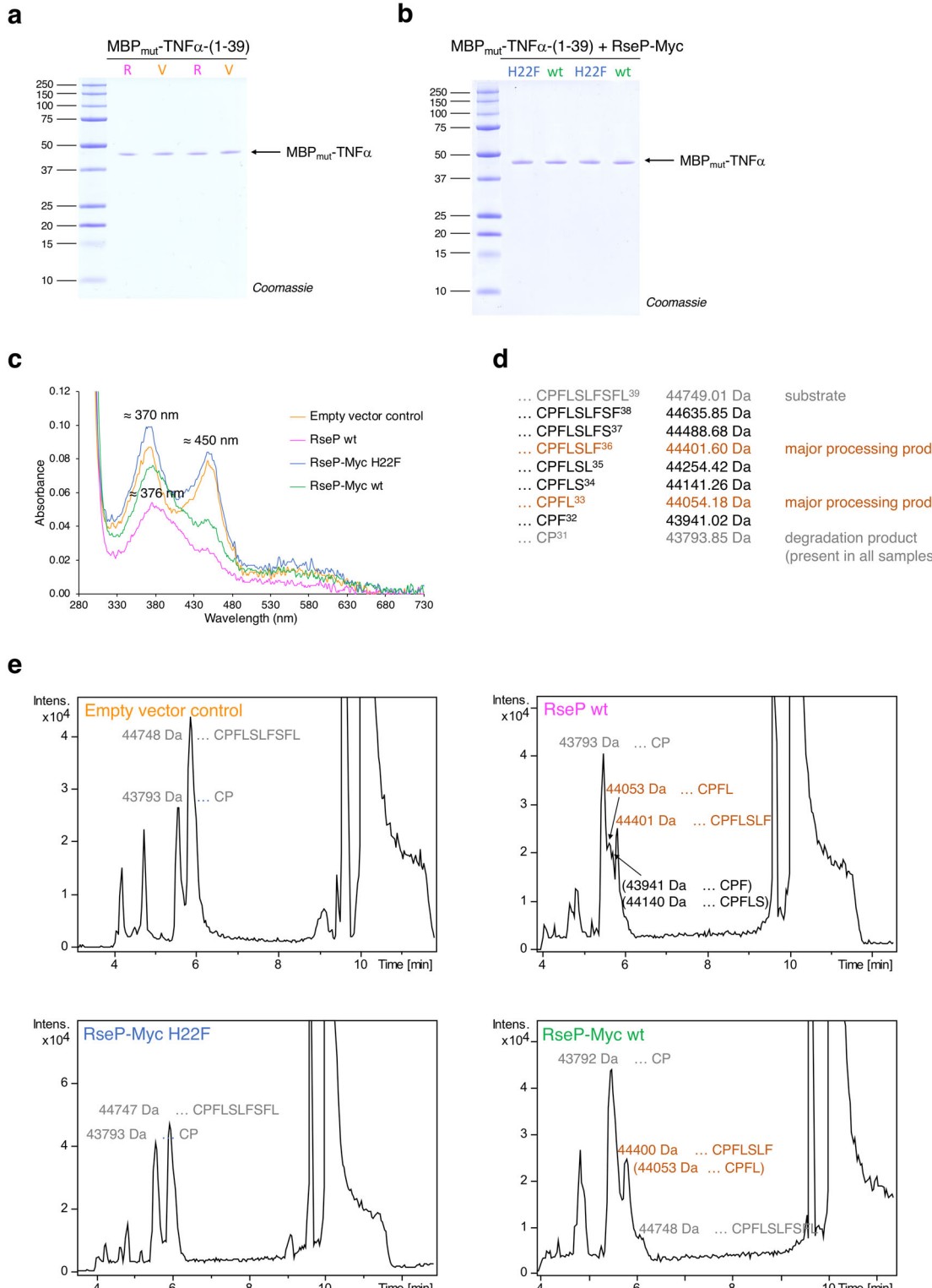

related to C-terminal truncation of the substrate. The mass spectrometric analysis suggests a processive cleavage mechanism of RseP and also gives insights into the specificity of RseP.

**Combining error-prone mutagenesis of RseP with iCliPSpy.** To identify residues in RseP that are critical for protease activity and/or substrate binding, we combined the in vivo assay iCliPSpy with error-prone mutagenesis of the *rseP* gene (Fig. 6a, b). We selected seven green *E. coli* clones co-producing an inactivated RseP

enzyme and the substrate MBP_{mut}-TNFα-(1-39)-L31P for sequencing of the *rseP* gene. In six out of seven clones, we found premature stop codons and frameshift mutations in addition to several amino acid exchanges (Fig. 6c), and therefore excluded them from further analysis. In contrast, the mutant designated RseP G1 contained eleven non-synonymous mutations, but neither premature stop codons nor frameshift mutations. We mapped the eleven amino acid exchanges onto the AlphaFold wt structure of RseP and found that the Y69H substitution is located

**Fig. 5 Determination of RseP cleavage sites. a** and **b** Membrane-associated MBP$_{mut}$-TNFα fusion proteins from IPTG-induced *E. coli* T7 Express cells were purified by amylose affinity chromatography and analyzed by SDS-PAGE. **a** *MBP$_{mut}$-TNFα-(1-39)-L31P* was co-expressed with pCOLADuet-1-*rseP wt* (R) or the empty vector control pCOLADuet-1 (V), or **b** with pCOLADuet-1-*rseP-Myc wt* (wt) or pCOLADuet-1-*rseP-Myc H22F* (H22F). Comparable amounts of purified proteins were run twice and alternately on SDS-PAGE to detect very small differences in molecular weight. **c** UV/Vis spectra of MBP$_{mut}$-TNFα fusion proteins purified from these different genetic backgrounds (orange trace, empty vector control; magenta trace, + RseP wt; blue trace, + RseP-Myc H22F; green trace, RseP-Myc wt). **d** Calculated average molecular weights of C-terminally truncated MBP$_{mut}$-TNFα-(1-39)-L31P fusion proteins (only the C-terminal residues of the MBP$_{mut}$-TNFα fusion proteins are shown here and in **e**). **e** Purified, putative unprocessed (left panels) and processed (right panels) MBP$_{mut}$-TNFα fusion proteins (compare **a** and **b**) were analyzed by LC-MS. The average molecular weights of the eluted MBP$_{mut}$-TNFα fusion proteins obtained by deconvolution of the ESI-MS spectra (compare Supplementary Fig. 4a-d) are given. The substrate MBP$_{mut}$- ... CPFLSFSFL[39] and the degradation product MBP$_{mut}$- ... CP[31] (found in all samples) are shown in grey, RseP major processing products MBP$_{mut}$- ... CPFLSLF[36] and MBP$_{mut}$- ... CPFL[33] in brown and additional processing products in black font. For MBP$_{mut}$-TNFα preparations from "Empty vector control" and "RseP-Myc H22F", we found a protein eluting at 9.1 min with an average molecular weight of about 45448 Da min, which was higher than expected for a MBP$_{mut}$-TNFα fusion protein and was therefore not further characterized. Compounds eluting at retention times above 9.5 min (corresponding to 90% acetonitrile) were contaminants not further characterized. **a** to **e** belong to one experiment, and the processing specificity of RseP wt was confirmed in this experiment by analysis of RseP-Myc wt. The C-terminal sequences of the substrate MBP$_{mut}$- ... CPFLSLFSFL[39] fusion protein and processing product MBP$_{mut}$- ... CPFLSLF[36] were confirmed by LC-MS/MS analysis (Supplementary Fig. 4e).

within the edge of the β-MRE at a position close to the active-site within the substrate binding groove (Fig. 6d). The edge structure was shown to be important for both protease activity and substrate binding[19,20]. For example, the Y69P substitution in RseP significantly impairs the cleavage of the model substrate HA-MBP-RseA148[19]. However, the introduction of a Pro residue can significantly alter the structure of the β-MRE[19,20]. To further investigate the role of Tyr69 in protease function without altering the overall structure of the β-MRE, we tested whether the single substitution Y69H, in which Tyr69 is exchanged for a more hydrophilic histidine residue, reduces RseP activity in vivo. Assuming that the color of colonies on agar plates reflects the proteolytic activity of RseP, we found a decreased in vivo activity of RseP Y69H compared to RseP wt (Fig. 6e), which was more pronounced for RseP G1 and the active-site variant RseP H22F (Fig. 6e). Next, we focused on a more detailed characterization of residues Tyr69 and Tyr428 by MD simulations and site-directed mutagenesis. Residue Tyr428 of TMD 4 was included in our studies because it is localized on the opposite side of Tyr69 at the putative substrate binding groove, and Tyr69 and Tyr428 together may work in gating substrates[22] into the binding groove (Fig. 6d).

**MD simulations of membrane-embedded RseP**. To further investigate the role of Tyr69 in the function of RseP, we performed MD simulations of RseP based on the structure predicted by AlphaFold[29,30] (with a Zn$^{2+}$ ion manually added to the active site), as the pdb file 7W6X for the crystal structure of full-length RseP[22] was not publicly available. To mimic the native environment, RseP was embedded in an artificial membrane resembling the lipid composition of *E. coli* membranes (see Supplementary Table 3 and Supplementary Note 3) and MD simulations (Fig. 6f and Supplementary Fig. 5a-e) were performed on a time scale of 2 microseconds. The structures obtained showed only minor changes in the RseP active site-containing membrane core compared to the AlphaFold prediction used for the simulations (and compared to the RseP crystal structure). However, the MD simulations indicated that the periplasmic tandem PDZ domain is flexible (Supplementary Fig. 5e). The MD simulations of membrane-embedded RseP predict that Tyr69 and Tyr428 are localized at the cytosol/membrane interface adjacent to the active site of RseP as part of the substrate binding groove (Fig. 6f and Supplementary Fig. 5). The physiological importance of a lipid molecule placed by MD simulation next to/in the substrate binding groove, however, is less clear. In the case of the MD simulation shown in Fig. 6f and Supplementary Fig. 5 this lipid was phosphatidylethanolamine (PE), in other simulations we found cardiolipin, suggesting that overall hydrophobicity is the

major determinant for insertion into the substrate binding pocket of RseP.

In our in vivo assay (Fig. 6g), the Y428H substitution had no significant impact on the proteolytic activity of RseP. Furthermore, the double substitution Y69H/Y482H was indistinguishable from the single substitution Y69H (Fig. 6g), suggesting that Tyr428 neither plays a role in binding the substrate MBP$_{mut}$-TNFα-(1-39)-L31P nor is important for catalytic activity. To support this result, MBP fusion proteins purified from *E. coli* cells co-overexpressing *MBP$_{mut}$-TNFα-(1-39)-L31P* with wt or mutant *rseP* genes were analyzed by SDS-PAGE (Supplementary Fig. 6a) and UV/Vis spectroscopy (Supplementary Fig. 6b). When *MBP$_{mut}$-TNFα-(1-39)-L31P* was co-expressed with *rseP Y69H* or with *rseP Y69H/Y428H*, heme remained bound to the reporter *via* bis-thiolate ligation (Supplementary Fig. 6b), confirming that intramembrane proteolysis by RseP is impaired when Tyr69 is exchanged for His. However, RseP Y428H behaved like the wt enzyme, processing the C-terminal residues of the reporter substrate (compare Fig. 5) and shifting heme binding towards a monomeric, thiolate-bound form.

**Discussion**

Recently, we observed that the N-terminal domain of TNFα comprising residues 1-39, TNFα-(1-39), dimerizes and binds heme *via* two cysteine residues, whereas the proteolytically processed peptide TNFα-(1-34) binds heme *via* one cysteine residue, as the dimerization motif S[34]XXS[37] within the TMD residues 31-39 of TNFα is disrupted[25]. In the present study, we exploit these spectroscopic properties of TNFα to develop a simple chromophore-based in vivo assay to monitor I-CliP activity. We chose *E. coli* RseP as a benchmark enzyme, which is well characterized by both in vitro and in vivo assays[23]. Our in vivo studies show that co-expression of *TNFα-(1-39)-L31P* with active *rseP wt* results in white *E. coli* colonies, while co-expression with inactive RseP H22F results in green colonies. After purification of the MBP-tagged reporter protein we confirmed by UV/Vis spectroscopy and mass spectrometry that TNFα-(1-39)-L31P is indeed cleaved by RseP. Mass spectrometric analysis showed that RseP engages in the trimming of MBP$_{mut}$-TNFα-(1-39)-L31P at L/F-S peptide bonds and releases the tripeptides S[37]FL[39] and S[34]LF[36], whereas the native substrate RseA is cleaved by RseP between Ala108 and Cys109[13]. However, the low substrate specificity of RseP and other I-CLiPs is well known[23,35]. Interestingly, tripeptide trimming is also observed for the processing of amyloid β-peptide 49 by γ-secretase, and in this case the trimming mechanism has recently been studied in detail using a novel peptide Gaussian accelerated molecular dynamics method[39]. In

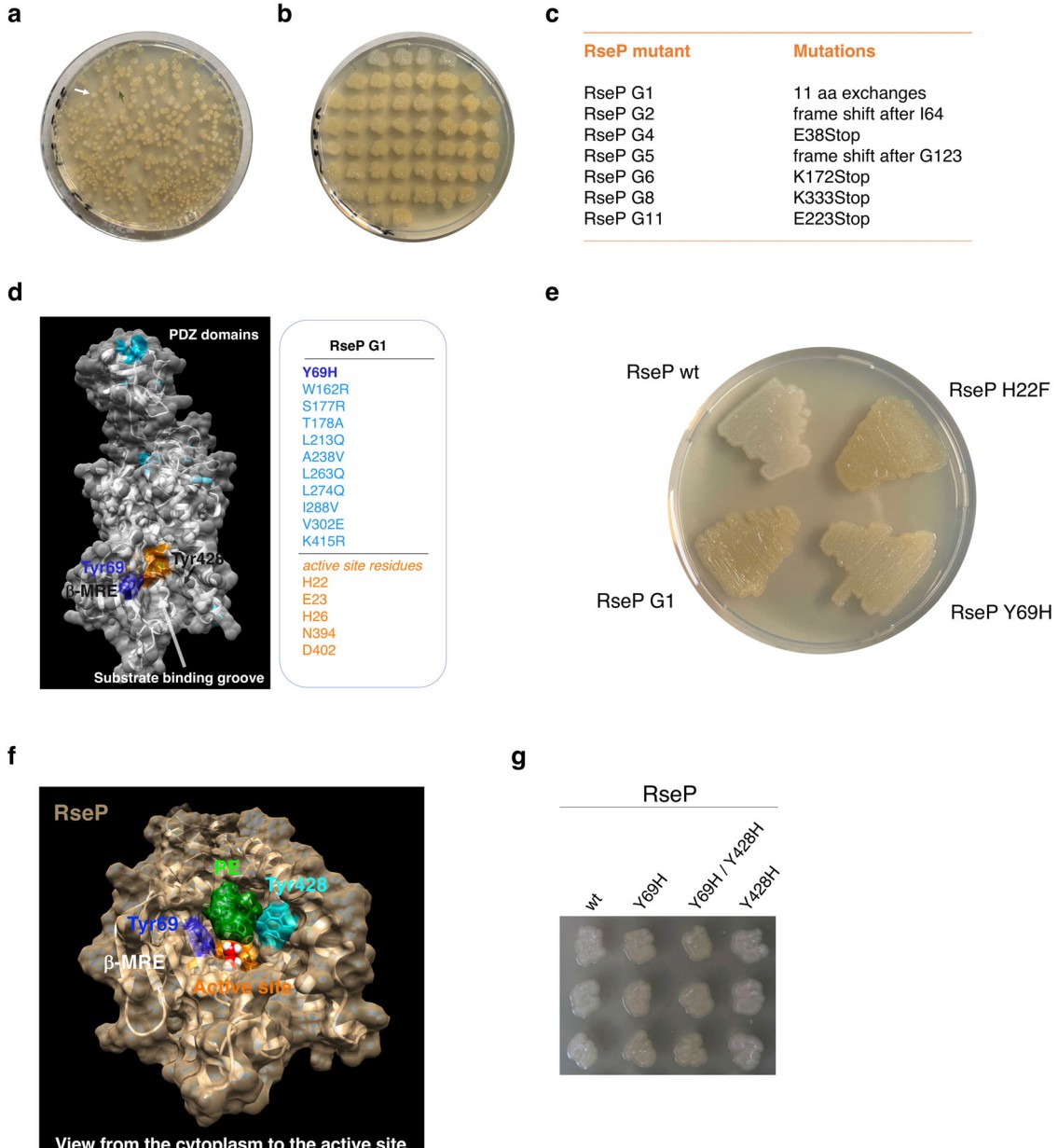

**Fig. 6 Error-prone mutagenesis of *rseP* in combination with the iCliPSpy assay. a** Mutant *rseP* genes were co-transformed with pETDuet-1-*MBP_{mut}*-*TNFα*-*(1-39)-L31P*. The transformation plates were incubated overnight at 37 °C and then at room temperature for two days (as an example, a white colony is indicated by a white arrow and an ocher-green clone by a green arrow). **b** White (presumably active RseP proteases; first row of the plate) and green clones (inactive RseP) were transferred to a new plate and incubated again for one day at room temperature. **c** From seven green clones expressing inactive RseP proteases, the plasmids pETDuet-1-*MBP_{mut}*-*TNFα*-*(1-39)-L31P* and pCOLADuet-1-*rseP* were purified together and the sequence of mutant *rseP* genes was determined. **d** The eleven non-synonymous amino acid substitutions in RseP G1 mutant were mapped to the AlphaFold structure of RseP wt (substituted amino acid residues are shown in light blue). Tyr69 (Y69, dark blue) is the only mutated residue located near the predicted active site (shown in orange); β-MRE, membrane-reentrant β-loop. **e** The Y69H substitution found in RseP G1 was introduced as a single mutation in RseP and analyzed together with RseP wt, RseP H22F or RseP G1 using the iCliPSpy assay. Cells were incubated at room temperature for four days (overnight incubation at 37 °C followed by incubation at room temperature resulted in less colored RseP Y69H cells). **f** MD simulations of the AlphaFold RseP wt structure (with Zn$^{2+}$ manually curated into the active site) in a membrane environment visualized with Chimera. View from the cytoplasm to the active site [composed of His22, His26 and Asp402 complexing the Zn$^{2+}$ ion (shown in black, but mostly hidden by two water molecules) and the residues Glu23 and Asn394] shown in orange. Tyr69 at the cytosol/membrane interface is shown in dark blue, Tyr428 in cyan; β-MRE, membrane-reentrant β-loop. Membrane lipids have been omitted (shown in Supplementary Fig. 5c), except for 1-hexadecanoyl-2-(9Z-hexadecenoyl)-*sn*-glycero-3-phosphoethanolamine (PYPE, in green), which is adjacent to/inside the binding groove. **g** Comparison of the in vivo activities of RseP wt, RseP Y69H, RseP Y69H/Y428H and RseP Y428H using the iCliPSpy assay. Cells shown in **g** had been incubated at room temperature for two days. Error-prone mutagenesis of *rseP* was performed once, in vivo analysis of RseP Y69H activity (**e**) was performed twice. In vivo assay for RseP wt, RseP Y69H, RseP Y69H/Y428H und RseP Y428H was repeated once with the same result as shown in **g**.

the future, MD simulations may also help to determine the mechanism of RseP more precisely and provide better insight into the mechanism of I-CLiPs in general.

The crystal structures of the tandem PDZ domain of RseP[16] and of the transmembrane core domain of an S2P metalloprotease from *Methanocaldococcus jannaschii*[40] led to an AlphaFold structure prediction[29,30] with very high confidence for RseP. We used this prediction for mapping mutations (Fig. 6d) and for MD simulations of RseP (Fig. 6f and Supplementary Fig. 5), because the pdb file for the crystal structure of RseP[22] was only available as we were working on the revised version of the manuscript but not in the beginning of our studies. Later, it also turned out that the AlphaFold structure prediction for the membrane core of RseP is very similar to the published crystal structure of RseP (Supplementary Fig. 5f), so that we did not repeat the MD simulations using the crystal structural data. Thus, our approach led to an overall correct picture of the structure of membrane-embedded RseP without knowledge of the crystal structure. Although the physiological importance of a lipid molecule placed next to/into the substrate binding groove by MD is unclear, this may be an important discovery as the lipid has to be exchanged for the substrate protein during catalysis. Future work will show whether this bound lipid is part of a lipid/substrate exchange during RseP-mediate cleavage of proteins.

Using the iCliPSpy in vivo assay, we identified the inactive RseP G1 mutant with eleven non-synonymous mutations, including the substitution Y69H, by error-prone mutagenesis. We focused on characterizing the single mutant RseP Y69H, which has not been previously described. Although the AlphaFold structures of RseP wt and RseP Y69H are nearly identical except for the Y69H exchange, we found decreased proteolytic cleavage of the substrate TNFα-(1-39) for RseP Y69H. The finding of decreased activity of RseP Y69H is consistent with the importance of the edge strand (residues Gly67 to Val70) for substrate binding by RseP, as it had been previously shown that proline substitutions of the edge residues Gly67, Gly68, Tyr69, and Val70 perturb the β-strand conformation of the β-MRE and abolish the proteolytic activity of RseP[19,20,22]. It is tempting to speculate that Tyr69 engages in a hydrophobic contact to a side chain of the unwound substrate peptide within the substrate binding groove and that the Y69H exchange reduces the affinity for the substrate. The crystal structure of RseP now shows that the bound inhibitor batimastat is in close contact to the side chains of Phe44 (which is part of the GFG motif[20]) and Tyr69. How the other mutations within the G1 mutant contribute to the impairment of RseP activity is less clear.

When the photoreactive hydrophobic amino acid, p-benzoyl-L-phenylalanine (pBPA) was incorporated at position 69, 71 or 74 of RseP (*e.g.* RseP Y69pBPA), RseP retained its proteolytic activity and the substrate RseA was photo-crosslinked to RseP, leading to a model in which the β-MRE binds to substrate TMDs and stabilizes their extended conformation[19,20,22]. In the recently proposed mechanism for substrate accommodation and cleavage in RseP, it is assumed that RseP is in equilibrium between a gate-open and gate-closed conformation, that the substrate is bound by the edge residues in the gate-open conformation, and that in a final step the transmembrane segment of the bound substrate is extended by the edge strand to promote proteolytic cleavage[22].

Although the obtained results were largely confirmative, our studies convincingly demonstrate the power of the combination of error-prone mutagenesis with the iCliPSpy assay for the characterization of RseP and possibly other I-CLiPs at the molecular level.

It is not known what fraction of the substrate protein MBP$_{mut}$-TNFα-(1-39)-L31P forms a dimeric complex with heme and what proportion is monomeric, since we neither know the heme-binding constant nor the equilibrium constant for (heme-promoted) dimerization of TNFα-(1-39). Also, the dimerization mechanism itself is not elucidated, but we favor a model in which heme is first bound by monomeric TNFα-(1-39) and then this protein-heme complex binds a second TNFα-(1-39) monomer assisted by the SXXS dimerization motif. We assume (considering the sizes of the peptide binding groove and the active-site) that monomeric TNFα-(1-39) (without ligated heme) and not the dimeric (heme binding) one is finally cleaved by RseP. This does not exclude that initially present dimeric TNFα-(1-39) is bound by RseP and that beta-strand addition dissociates the dimer before cleavage. RseP-catalyzed cleavage of the dimerization motif within monomeric TNFα-(1-39) will result in decreased amounts of peptide TNFα-(1-39) available for (heme promoted) dimerization and increased amounts of cleaved peptides binding heme by only one Cys residue.

In summary, we present iCLiPSpy, a chromophore-based assay for in vivo analysis of intramembrane proteolysis. In addition, combining iCliPSpy with error-prone mutagenesis can help in targeted protein design, for example to increase protease activity. In principle, iCLiPSpy can be extended to other I-CLiPs (that cleave type II transmembrane proteins) as well as to other protein substrates of the THOMAS (**T**hiolate-**H**eme **O**ligomeric type II trans-**M**embrane proteins with heme binding mode **a**djusted by **S**PPL2a/b) protein family[25]. For example, CD74-NTF also binds heme *via* bis-thiolate ligation[25] and can be used to expand the repertoire of substrates to study I-CLiPs with different substrate specificities. The investigation of other I-CLiPs, such as SPPL2a/b proteases, using iCliPSpy depends on their expression in *E. coli* and moreover, in vivo detection of heme binding substrate proteins such as CD74-NTF is only possible at high expression levels (changing the color of cells). Further research and development is needed to transfer this approach to an eukaryotic expression system such as yeast. In combination with the over-expression of I-CliP reporters such as TNFα-(1-39)-L31P or CD74-NTF, green yeast colonies would turn into colorless colonies due to the activity of I-CLiPs. Such a system would be adaptable for high-throughput screening approaches aimed at discovering inhibitors or activators of I-CLiPs, which is of great interest for proteases being explored as biomedical targets.

In pathogenic bacteria, site-2 proteases play an important role in sensing host signals and regulating the expression of virulence genes during infection[41]. In *Vibrio cholerae*, for example the I-CLiP YaeL (a homolog of RseP) is involved in the inactivation of the membrane localized transcription factor TcpP *via* a two-step RIP pathway. TcpP activates transcription of *toxT*, which encodes the direct activator of toxin and pilus genes[42]. In *Streptococcus pneumoniae*, RseP cleaves the competence protein ComM to control its quantity. ComM protects *S. pneumoniae* from CbpD-mediated lysis during competence[43,44]. The iCLiPSpy assay can provide a simple yet powerful tool for the identification/detection and characterization of these and other I-CLiPs from pathogenic bacteria and opens up the possibility of searching for small molecule modulators of protease activity in an in vivo model system.

## Methods

**Expression plasmids**. MBP$_{mut}$-TNFα-(1-39)-L31P, MBP$_{mut}$-TNFα-(1-34)-L31P, His-MBP-TNFα-(1-39)-L31P and *E. coli rseP wt, rseP-Myc wt* and mutant genes, respectively, optimized for expression in *E. coli* were synthesized by Invitrogen GeneArt (Thermo Fisher Scientific). MBP$_{mut}$ contains substitutions increasing the affinity for amylose (A312I, I317V), reducing the surface entropy (D82A, K83A, E172A, N173A, and K239A) and increasing the crystal-packing interactions between neighboring MBP molecules (A215H/K219H)[31]. MBP only contains the substitution A312V (numbering of amino acid residues refers to the sequence of periplasmic MBP lacking the signal peptide). To minimize differences in expression rates of *rseP wt, rseP H22F, rseP Y69H, rseP Y69H/Y428H, rseP Y428H* and *rseP-*

*Myc wt, rseP-Myc* H22F genes due to different codon usage, the sequences only differ in the exchanged codons. Cloning into *NcoI/XhoI* sites of expression vectors pETDuet-1 (ColE1 replicon, ampicillin resistance) and pCOLADuet-1 (COLA replicon, kanamycin resistance; compatible with pETDuet-1), respectively, was performed using standard molecular biology techniques. The DNA sequence of all cloned genes was confirmed using appropriate primers. DNA sequences of all constructed plasmids (listed in Supplementary Table 1) are given in Supplementary Note 1, so that the exact amino acid sequences of all proteins used in this study are also available. Due to the cloning procedure the second residue (Leu; codon CTG) of all RseP variants is exchanged for Val (codon GTG). RseP L2V and RseP L2V H22F (and so on) are referred to as RseP wt and RseP H22F, respectively, in this paper. Transformation and expression were performed with NEB (New England Biolabs) 5α and T7 Express *E. coli* cells.

**In vivo cleavage of MBP_mut-TNFα-(1-39)-L31P by RseP**. T7 Express *E. coli* cells co-transformed with pTK1130 [= pETDuet-1-*MBP_mut-TNFα-(1-39)-L31P*] and pCOLADuet-1 encoded *rseP* genes were streaked from bacterial glycerol stocks onto selection plates [LB agar substituted with 2 g glucose l-1, ampicillin (100 μg ml-1) and kanamycin (50 μg ml-1), but containing no δ-aminolevulinic acid hydrochloride[45], incubated overnight at room temperature or 37 °C. Photos of the plates were taken after incubation for several additional days at room temperature. Expression of *MBP_mut-TNFα-(1-39)-L31P* and *rseP* occurred because the T7 expression system is leaky.

**Purification of membrane-associated MBP fusion proteins**. As recently described, we used the MBP tag for amylose affinity purification of the MBP-TNFα fusions in combination with n-dodecyl-β-D-maltopyranoside (DDM, Avanti) as detergent for membrane protein solubilization and subsequent chromatography[25]. *E. coli* T7 Express *E. coli* cells transformed with the appropriate MBP-TNFα encoding pETDuet-1 plasmids (see Supplementary Table 1) were grown in LB medium substituted with 2 g glucose l-1 and ampicillin (100 μg ml-1) at 37 °C. Co-expression experiments of pTK1130 [= pETDuet-1-*MBP_mut-TNFα-(1-39)-L31P*] and pCOLADuet-1 encoded *rseP* genes, were also performed in *E. coli* T7 Express cells. Cells were grown in this case in LB medium substituted with 2 g glucose l-1, ampicillin (100 μg ml-1) and kanamycin (50 μg ml-1). For both kinds of experiments, over-expression was induced with 50 μM isopropyl 1-thio-β-D-galacto-pyranoside (IPTG, Sigma-Aldrich) at an $A_{600}$ of about 0.3, and cultures were then further shaken at 37 °C for about 2 h. Cells from a one liter culture were harvested, resuspended in 30 ml TN buffer (50 mM Tris/HCl pH 7.4 at room temperature and 150 mM NaCl) containing 1 mM phenylmethylsulfonyl fluoride (PMSF, Sigma-Aldrich), frozen in liquid nitrogen and stored at -80 °C. For the experiments shown in Figs. 2d-f, 4, and Supplementary Fig. 3, 0.6 mM δ-aminolevulinic acid hydrochloride was added before induction with IPTG[45]. However, δ-aminolevulinic acid hydrochloride was not used in all other experiments. Thawed cells were again substituted with 1 mM PMSF and then disrupted by sonication. The membrane pellet obtained after 30 to 45 min centrifugation at 40,000x *g* and 4 °C was resuspended in 15 ml TN buffer containing 2% of the detergent DDM. Then, membrane proteins were solubilized at 4 °C for about 30 to 45 min in a 50 ml tube using a tube roller mixer. The supernatant obtained after centrifugation for 30 min at 40,000x *g* and 4 °C was subjected to amylose (NEB) affinity chromatography at room temperature. Amylose affinity columns were equilibrated and washed with TN buffer containing 0.2% DDM. MBP fusion proteins were eluted with 10 mM maltose (Sigma-Aldrich) in equilibration buffer and UV/Vis spectra of the eluted protein solutions were recorded at room temperature in a 1 cm Hellma quartz absorption cuvette using the NanoDrop 2000c spectrophotometer from Thermo Fisher Scientific with elution buffer as blank[25].

**Gel filtration**. For gel filtration experiments, eluted MBP fusion proteins were concentrated using Vivaspin 500 columns at 4 °C and then stored at 4 °C. Gel filtration was carried out as described[25] at 4 °C using an ÄKTA pure system from GE Healthcare and a Superose 6 Increase 10/300 GL column for high-resolution size exclusion chromatography. 100 μl of protein aliquots were subjected to the column equilibrated in TN buffer containing 0.2% DDM at a flow rate of 0.4 ml min-1. The elution was followed by absorbance at 280 nm and simultaneously for the detection of heme at additional wavelengths (such as 371 nm and 451 nm), and 0.4 ml fractions were collected. The void volume of the used column was determined with Blue dextran 2000 (GE Healthcare) to be 8.5 ml.

**Mass spectrometric analysis of RseP processing products**. Intact masses of MBP-TNFα fusion proteins were determined using reversed-phase (RP) ultra high-performance liquid chromatography (UHPLC) coupled to electrospray ionization mass spectrometry (RP-ESI-MS). Briefly, RP separation was performed on a 1290 UHPLC (Agilent Technologies) equipped with a PLRP-S column (1.0 ×50 mm, 5 μm, 1000 Å; Agilent Technologies) at 80 °C and 0.35 ml/min. Solvent A was 0.1% formic acid (FA) in water. Solvent B was 0.1% FA in acetonitrile. Total run time was 17 min and the gradient was as follows. 0-0.5 min: 20% B, 0.5-8.5 min: linear increase from 20% to 50% B, 8.5-9.0 min: linear increase from 50% to 90% B, 9.0-11.0 min: 90% B, 11.0-11.5 min: linear decrease from 90% to 20% B, and 11.5-17 min: re-equilibration at 20% B. MS analysis was performed (min 4 to 13) in

positive polarity scan mode on an Impact II ESI-QqTOF mass spectrometer (Bruker Daltonics), in a mass range from m/z 600 to 7000. Following ESI source parameters were used: end plate offset 500 V, capillary voltage 4500 V, nebulizer gas 3.0 bar, dry gas 12.0 l/min, dry temperature 300 °C. High mass calibration was performed using cesium fluoroheptanoate in enhanced quadratic calibration mode. Mass spectra were processed in DataAnalysis software version 4.4 (Bruker Daltonics). In brief, the protein is getting detected with different charge states in parallel, that are then deconvoluted to obtain the molecular weight of the protein. Masses were determined using maximum entropy charge deconvolution in a range of 10,000 – 50,000 Da. To confirm RseP cleavage specificity in a complementary analysis, tryptic peptides of MBP-TNFα fusion proteins were sequenced by LC-MS/MS (for a detailed setup of this peptide analysis see Supplementary Note 2). Since no extracted ion chromatograms (XIC) of the intact mass analysis could be obtained due to the nature of the analysis, quantification was performed by LC-MS/MS on the peptide level (compare Supplementary Table 2).

**Error-prone PCR**. Error-prone PCR[28] was performed using the JBS (Jena Bioscience) Error-prone PCR kit with plasmid pTK1121 (= pCola-Duet-1-*rseP* wt) as template for the *rseP* gene and using the forward and reverse PCR primers TK154 (5′-AACTTTAATAAGGGAGATATACCATGG-3′) and TK155 (5′-GGTTTCTTT ACCAGACTCGAGTTATTA-3′) (*NcoI* and *XhoI* restriction sites are underlined). The rate of mutagenesis of this method is in the range of 0.6-2.0% and the mutations are randomly distributed throughout the amplified PCR product[28]. The PCR product was purified using the Qiagen PCR purification kit, double digested with *NcoI* and *XhoI* and then isolated by agarose gel electrophoresis using the Qiagen gel extraction kit. The *NcoI/XhoI rseP* (error prone) PCR fragment was then ligated with *NcoI/XhoI* double-digested pCOLA-Duet-1 and the ligation mixture was transformed into *E. coli* NEB5α cells. An aliquot of the transformation mixture was used to inoculate 50 ml LB + glucose medium. Plasmids of cells grown overnight at 37 °C were then purified using the Nucleobond PC100 Midi kit. The purified plasmid pCola-Duet-1-*rseP* (error prone) was subsequently co-transformed with pTK1130 [= pETDuet-1-*MBP_mut-TNFα-(1-39)-L31P*] into NEB T7 Express *E. coli* cells. The diluted transformation mixture was spread onto selection plates containing ampicillin and kanamycin, incubated overnight at 37 °C and then for two days at room temperature. White and green clones were then transferred to a fresh selection plate, incubated overnight at 37 °C and then for one additional day at room temperature. Plasmid DNA [pETDuet-1-*MBP_mut-TNFα-(1-39)-L31P* and pCola-Duet-1-*rseP* (error prone)] of green clones was purified and the sequence of the mutated *rseP* genes was determined using the pCOLA-Duet-1 specific primers TK160 (5′-CTCCTGCATTAGGAAATTAATACG-3′) and TK161 (5′-GTGCTTC TCAAATGCCTGAGG-3′) and in addition by using several different *rseP* specific primers.

**SDS-PAGE and immunoblotting**. As described[25], proteins were separated using Tricine-SDS-polyacrylamide (10%) gel electrophoresis under reducing (1x sample buffer containing 10 mM dithiothreitol, DTT) conditions[46]. Molecular weights of standard proteins (all blue prestained protein standard, Bio-Rad) are indicated on the side of the gel pictures. For documentation, destained gels were scanned using an Epson Perfection V850 Pro scanner with 1200 d.p.i. resolution. Proteins were electrophoretically transferred to PVDF membrane (Immobilon-FL, Merck Millipore) after SDS-PAGE by semi-dry blotting. Myc-tagged proteins were detected with a monoclonal anti-Myc Tag antibody (Sigma-Aldrich SAB1305535, clone 9E10, diluted 1:500) as first antibody, an Alexa 680 goat-anti-mouse antibody (Life Technologies, Thermo Fischer Scientific, 0.8 μg ml-1) as secondary antibody and then using the Odyssey Imaging System (Li-Cor Biosciences).

**AlphaFold and MD simulations**. We used the AlphaFold[29,30] predicted structure of *E. coli* K12 RseP (with a Val residue in position 2) for our MD simulations. Into the predicated structure we manually placed a $Zn^{2+}$ ion coordinated by the side chains of residues H22, H26 and D402. The modified structure was then embedded in silico into a described *E. coli* model membrane system containing about 75% cardiolipin and about 25% PE with multiple types of acyl chains[47] using the CHARMM GUI membrane builder interface[48,49]. Six MD simulations for membrane-embedded RseP wt, were performed each for 2 μs using GROMACS[50] software and CHARMM36m forcefield[51] (for a detailed setup of the methodology compare Supplementary Table 3 and Supplementary Note 3). To choose the statistically most representative structures for illustration, the obtained trajectories were analyzed by clustering using the GROMOS method[52] applying an RMSD cutoff of 0.15 nm. All pdb files were visualized with UCSF Chimera[53].

**Statistics and reproducibility**. No statistical analyses were performed. Wt and mutant proteins were always purified in parallel using the same purification procedure including the same buffers and detergent solutions (and using cells grown under the same conditions). The precise numbers of purifications are indicated in the figure legends.

**Reporting summary**. Further information on research design is available in the Nature Portfolio Reporting Summary linked to this article.

## Data availability

All underlying data of this study are available in the main article and Supplementary Information. The source data behind Figs. 2f, 4c-e, 5c are provided as Supplementary Data 1 and the DNA sequences of all constructed plasmids are included in Supplementary Note 1. The uncropped version of Fig. 2c is included as Supplementary Fig. 7. The mass spectrometry proteomics data have been deposited with the ProteomeXchange Consortium via the PRIDE[54] partner repository with the dataset identifier PXD038785 and 10.6019/PXD038785. All other data are available from the corresponding authors upon reasonable request.

## Code availability

No customed or in-house software was used in the analysis of the MD simulations. AlphaFold2 model and output files as well as MD simulations parameter files are publicly available at Zenodo: https://doi.org/10.5281/zenodo.7612846[55].

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

## Acknowledgements

This work was funded by the Deutsche Forschungsgemeinschaft (DFG, German Research Foundation) – Project Number 112927078-TRR83, and by the Ministry of Education and Research (BMBF) – project Drugs4Future (13FH8I05IA) within the "FH-Impuls" framework. CSC-IT Centre for Science Ltd. (Espoo, Finland) is acknowledged for excellent computational resources. The authors gratefully acknowledge the data storage service SDS@hd supported by the Ministry of Science, Research, and the Arts Baden-Württemberg (MWK) and the DFG through grant INST 35/1314-1 FUGG and INST 35/1503-1 FUGG. For the publication fee we acknowledge financial support by DFG within the funding programme „Open Access Publikationskosten" as well as by Heidelberg University.

## Author contributions

T.K. formed the concept for the in vivo intramembrane-cleaving protease assay, designed the overall research in coordination with B.B. and performed all experiments apart from analytical mass spectrometry which was done by R.G., F.R. and C.H. R.B. and F.L. did the MD simulations. T.K. prepared Figs. 1 to 6 and Supplementary Figs. 1 to 7 (apart from Supplementary Fig. 5e which was prepared by RB). Funding acquisition: B.B., C.H. and W.N. T.K. and B.B. wrote the manuscript. Review and editing: R.G., F.R., F.L., R.B., C.H. and W.N.

## Funding

## Competing interests

The authors declare no competing interests.
