## [Peer Review File · Communications Biology]

Reviewers' comments:

Reviewer #1 (Remarks to the Author):

The authors report a novel assay to study regulated intramembrane proteolysis (RIP). In prior work, the authors showed that RIP of TNF α changes its heme-binding characteristics. Here, they build on the previous work to create an innovative assay in which RIP changes the colony color of *E. coli* engineered to co-express an MBP-TNF α substrate (which is green due to high-affinity bis-thiolate ligation of heme) and the intramembrane metalloprotease RseP. Active RseP changes the colony color from light green to white (due to low-affinity penta-coordinated heme binding by the cleaved MBP-TNF α), whereas inactive RseP fails to change the colony color.

The breadth of applicability of the assay remains to be established. TNF α is normally cleaved by a signal peptide peptidase-like intramembrane aspartyl protease in eukaryotic cells. RseP cleaves the anti-sigma factor RseA, remnant signal peptides, and other substrates in *E. coli*. Therefore, the applicability of the assay may be broad, including (at least) the many intramembrane proteases that cleave Type 2 membrane proteins (with a single transmembrane segment and an extracytoplasmic C-terminus).

Importantly, the assay is amenable to high-throughput screening, which the authors demonstrate by screening *rseP* mutants generated by error-prone PCR. Screening of compound libraries for modulators of intramembrane proteases (that work in the assay) should be possible. High-throughput screening of compound libraries and intramembrane protease substitutions represent significant potential advances for the intramembrane protease field.

In general, the data are convincing and presented well in the manuscript. The colony-based assay is supported by SDS-PAGE of whole-cell extracts and by purification of MBP-TNF α substrate and product followed by spectroscopic and gel filtration/spectroscopic analyses. However, Figure 4 needs clarification and possibly additional experimental work, and the last section of the Results and Figure 6 add little to the manuscript and should be moved to the Supplemental Material.

I support publication after modifications to address the following comments.

Major comments:

p. 6, l. 9 – do the authors have evidence that the shortened linker peptide increases the stability of the fusion protein? If so, the evidence should be shown. If not, or if only the protein accumulation was measured, then the statement should be revised accordingly.

p. 7, l. 6 – “product of I-CliP-mediated proteolysis in *E. coli*” is incorrect. Rather, the authors expressed MBPmut-TNF α -(1-34)-L31P, which was designed to mimic the predicted cleavage product.

p. 8, l. 5 from bottom to the end of the section – the presentation of Figure 4b-d is oversimplified and unclear. For 4b, “slightly smaller apparent molecular weight” is oversimplified since only the migration of the leading edge of the fusion protein bands differ and for the most part their migration overlaps. SDS-PAGE at higher resolution with less protein may resolve the cleavage product(s) from the uncleaved substrate in the RseP wt sample. The authors also ignore the presence of uncleaved substrate in the RseP wt sample in their presentation of 4c, yet the lower panel of 4d shows that dimer is present; however, it is unclear why the dimer elutes slightly later than the dimer in the RseP H22F sample (upper panel of 4d). The result seems to suggest that the RseP wt sample contains species with C-termini between residues 34 and 39, which form dimers. Also unclear is which fractions were subjected to UV/VIS spectroscopy in 4e. The figure legend states “fractions containing heme binding MBPmut-TNF α fusion proteins”, suggesting that the spectrum of the RseP wt sample is the

fractions containing the dimer peak, but it seems more likely to be the monomer peak, given the shift to 392 nm. This needs to be clarified.

Related to the preceding comment, a weakness of the manuscript is the incomplete characterization of the RseP cleavage product(s). Did the author attempt accurate mass determination of intact and/or trypsin-digested purified product(s)?

p. 10, l. 20 – the authors fail to comment on the results shown in Figure 6b. The Y69H and Y69H/Y482H variants of RseP are expected to produce products that migrate more slowly than those from wt and Y482H, but this is not convincing in Figure 6b. The band from Y69H/Y482H appears to migrate a bit more slowly, but less fusion protein is present, which may affect the appearance of the band. As noted above for Figure 4b, SDS-PAGE at higher resolution with less protein (and an equal amount of each protein) may resolve the fusion proteins in this case too. As noted above, accurate mass determination of intact and/or trypsin-digested purified product(s) may help.

Related to the preceding comment, the last section of the Results adds little to the manuscript. As mentioned in the Discussion, it was shown previously that RseP Y69P is severely impaired for proteolytic activity, so it is not surprising that Y69H does as well. The Y482H substitution in RseP has little effect. The AlphaFold prediction looks similar to the crystal structure depictions in preprint reference 20. The purpose of the MD simulations is not explained, no strong conclusions are made, and the significance of a lipid in the substrate binding groove is unclear. The last section of the Results and Figure 6 should be moved to the Supplemental Material.

p. 13, l. 12 – why do the authors assume that monomeric TNFalpha-(1-39) is bound by RseP and that the dimeric peptide is not bound? It seems conceivable that RseP binds to the dimer initially and then beta-strand addition dissociates the dimer, preventing high-affinity bis-thiolate ligation of heme and allowing cleavage of a monomer.

p. 13, l. 19-23 – the authors should expand their discussion of the applicability of iCLiPSpy to other intramembrane proteases. For example, RseP has served as a model for a large family of PDZ domain-containing intramembrane metalloprotease found in nearly all bacteria, including many important pathogens where they affect expression of virulence genes.

Minor points:

p. 2, l. 5 – change “belong to the most” to “are” since “most” is debatable

p. 3, l. 2 – change “proteolytically” to “proteolytic”

p. 3, l. 10 – define “RIP” here or earlier

p. 3, l. 3 from bottom – the GFG motif of RseP is also implicated in substrate binding, which should be mentioned and reference 33 should be cited

p. 4, l. 1 – the original reference to RseP cleavage of signal peptides should also be cited (Saito et al. 2011 PNAS 108:13740)

p. 4, l. 4 – “spectroscopically” should be “spectroscopic”

p. 4, l. 8 – define “NTF”

p. 6, l. 2 – change “putative” to “potential” since the authors hypothesize that TNFalpha-(1-39) may be a substrate, but no prior evidence supports this

p. 6, l. 17 – change “leaking” to “leaky expression” (both occurrences)

p. 6, l. 23 – suggest changing “uncolored” to “uncolored / white” here and using “white” thereafter to avoid confusion; change “scratched” to “scraped” here and elsewhere

p. 7, l. 12 – change “recently published” to “published previously” since one of the papers cited was published in 2004

p. 7, l. 16 – reference 33 should also be cited here

p. 10, l.11 – “site” should be “side”

p. 12, l. 4 – cite a reference at the end of the sentence

p. 13, l. 2 from bottom – “high-screening” should be “high-throughput screening”

p. 14, l. 3 – “mutations” should be “substitutions” here and elsewhere when describing amino acid changes

p. 14, l. 7 – a single exchanged codon can alter the expression of a protein, so the sentence is incorrect. Are the authors indicating that genes were sequenced to ensure there were no other mutations?

p. 14 – define TN buffer

p. 15 – change “was now embedded” to “was then embedded”

p. 22 – The y-axes of the graphs in Figure 4d should be explained (relative absorbance?)

p. 30 – Figure 5a – arrows pointing to examples of one white colony and one green colony on each plate would be helpful since the difference is subtle

Reviewer #2 (Remarks to the Author):

Dear Editor, dear Authors,

The submitted paper “I-CliPSpy, a novel heme binding tag-based assay to characterize structure and function of the bacterial intramembrane protease RseP in vivo” describes a new assay to test the activity of intramembrane proteases in vivo in E.coli cells. Thomas Kupke and colleagues nicely show that a heme binding intramembrane substrate based on TNF α changes color upon cleavage by the target protease RseP. Protease and Substrate were co-expressed in E.coli cells. Overall, the study is conclusive, the experiments are described very well and were performed in a very reasonable and intuitive way. However, some minor issues need to be addressed before publication.

1. One of the reporter constructs that the authors used carried an N-terminal His10-tag (construct 2). Importantly, this construct resulted in only weakly colored cells (Fig. 2a), which is a very remarkable finding. However, the authors do not provide an explanation why they used this construct and why the cells were only weakly colored. An explanation would be very valuable. One could guess, based on Fig. 2c, color is reduced because of a reduced expression yield.

2. In Figure 4d the authors show that on a SEC column the dimer peak of the reporter shifts to the right (lower molecular weight) upon cleavage by RseP wt. According to the authors hypothesis, the cleavage product should separate into monomers after cleavage, because of upon cleavage the dimerization motif is removed and thus heme promoted dimerization is prevented. I can understand that the monomer peak gets more prominent when RseP wt was used as compared to RseP H22F. How can this shift of the dimer be explained? And how does it fit to the assumption that RseP wt is cleaving monomeric TNF-reporter, as mentioned in the discussion on page 13?

3. In the discussion the authors write that they assume that RseP binds the monomeric and not the dimeric heme binding peptide. However, to me it is unclear how they come to this assumption. I think it would be good if the authors could explain this in more detail.

4. The authors did not mention whether it would be possible to do a quantitative analysis of the enzymatic activity using their assay. They suggest though that the assay may be used for screening of ligands, inhibitors or mutants of intramembrane proteases. However, for these screening applications it is important to have a quantitative read out. The “greenish” or white color of the E.coli colonies has to be measured by an instrument. How can such a quantitative readout be implemented? Maybe the reduction in the 450 nm peak could be used for quantification? It would help if the authors could show a quantitative measurement of the E.coli cell coloring, at least for one of their experiments.

5. The manuscript could benefit if the cleavage of the TNF-reporter would also be cross-confirmed in another experiment. Mass spectrometry could be a straight forward method to go for. Or, alternatively, adding a C-terminal Tag (e.g. strep-tag), which would allow to do western blot

experiments. A C-terminal tag might however be incompatible with the cleavage by the target protease.

Minor issues:

6. The authors write that a crystal structure of RseP was determined. However, to interpret their results they use an AlphaFold generated model of RseP, rather than the crystal structure. Only in the discussion they mention that they do so because the crystal structure was not yet released by the wwPDB. This information is important and should be included in the results section, where the AlphaFold model is presented.

7. The authors used the detergent DDM to solubilize the membrane protein. However, this was only explained in the legend of Figure 2, not in the main text. I would recommend including the explanation for the DDM in the main text.

8. The authors suggest that their method can also be used as a screening assay for other intramembrane proteases. I'd recommend mentioning some specific examples of proteases that could be potential targets for this assay.

9. Overall, the manuscript is clearly written. It is easy to follow the content of the manuscript. However, the authors use numerous abbreviations. This is of course ok, but some of the abbreviations are not explained upon first use and some are not explained at all (e.g. RIP, TMD, ICD). Scientist working within this specific field might be familiar with these abbreviations. However, it will be hard for a non-specialist to understand the content of the paper if some important abbreviations are not explained.

10. On page 14 it says "... TN buffer containing 2% of the detergent DDM (FROM)." I guess this is a typo.

11. The authors provide sufficient details in the Material & Methods section and in the Supplement so the experiments can be certainly reproduced. The only thing I can criticize is that the composition of the TN buffer was not specified. At least I could not find it.

12. Figure 5c: The orange color shows the location of active site residues. However, to me it is unclear where Tyr428 is exactly localized there. I'd recommend to e.g. label Tyr428 in another color or by an arrow.

13. Figure 6: The labeling of panels e, f and g is hard to read. I'd recommend increasing the contrast of the labeling.

Overall, the manuscript and the figures are clear and well-made and I believe that this article will be read with great interest by the scientific community.

Reviewer #3 (Remarks to the Author):

Kupke et al. present a novel *in vivo* assay for the activity of intramembrane metalloprotease RseP that they term I-ClipSpy. The assay is based on an observation by the authors made earlier. The activity readout is indirect, and it is based on the transmembrane helix fragments of TNF produced by SPPL2a/b binds heme via thiolate or bis-thiolate ligation, where the two resulting types of complexes have different spectral properties. The reporter is designed such that it binds heme by the bis-thiolate ligation in the uncleaved state, while by the (single) thiolate mode in the RseP-cleaved state. Since the two states differ in their binding to heme and have different spectral properties (bis-thiolate

adduct is green, while the thiolate one is colourless), heme binding and spectral (colour) changes report on RseP activity. The authors demonstrate the application of the assay by randomly mutagenizing RseP and isolating an inactive mutant with 11 mutations. Based on this analysis and investigation of the AlphaFold2 model of RseP they pinpoint the Y69 residue as participating in substrate binding. Finally, the authors propose that this assay can find general use for testing RseP inhibitors and activators, and for investigating other intramembrane proteases. This is a novel assay, but it should be characterized better, which would also help clarify how likely it is that it would find a more general use.

I have the following criticisms and remarks to this study.

Major points

1. The membrane topology of the reporter is not entirely clear from the description in the paper. Since this is an engineered protein, it has to be properly described. Wild type TNF is a type II membrane protein, hence its C-terminus faces extracellular space. If signal peptide precedes MBP, TNF TMD would end up as a type I TMD, with its cleavage site region by RseP located at the intracellular side of the membrane. The MBP-based reporter could however also adopt a type II topology, dictated by the TNF transmembrane domain. Unfortunately, the exact design of the reporter is impossible to infer from the current description in the manuscript. Please add clearly the information on whether MBP contains its signal peptide, and specify the sequence and TMD boundaries of the TNF fragment. Clearly indicate this in Fig.1 and in the text/methods.
2. Following on from point 1: is the reporter indeed membrane-localised? The efficiency of membrane targeting should be analysed by cellular fractionation (cytosol/membrane) and tightness of membrane association should be analysed (peripheral vs transmembrane) by sensitivity to high salt washes and detergent extraction. The topology of the reporter can be analysed experimentally by protease protection assay, which should probably be undertaken to characterize this novel reporter rigorously.
3. Cleavage site by RseP should be located within the TMD of TNF. How will heme bind to this intramembrane segment of TNF?
4. It seems that the assay employs endogenous heme from E.coli whose synthesis is stimulated by the δ -aminolevulinic acid in the growth medium. It must be made clearer in the manuscript. There is no mention of the growth medium containing δ -aminolevulinic acid) within the Methods.
5. The cleavage site(s) in the TNF part of the reporter by RseP are only assumed based on the spectral properties of the heme conjugates and similarity to the SPPL2a/b ones. Cleavage site position(s) by RseP in the membrane localized reporter must be demonstrated by mass spectrometry.
6. The expression levels of the wild type and mutant RseP variants (e.g. in Fig. 3, Fig. 6) must be shown by immunoblot.
7. How do the chemical identity, and TNF-peptide binding and spectral properties of the human heme (presumably used in the ref. 22 often referred to) and the E.coli heme (presumably used here) relate?
8. Depending on the above details, how can this assay be generalized to other intramembrane proteases? It is based on using TNF and it is not clear if and which intramembrane proteases are likely to cleave TNF and in the expected topology. Can the heme binding peptide be engineered into other intramembrane protease substrates such that the heme binding readout can be used?
9. How do the authors envisage that screening for inhibitors and activators can be accomplished with sufficient sensitivity and quantifiability in their current assay format? To what extent is the colony colour proportional to RseP activity?
10. The speculations on the role of Tyr69 in substrate interaction would deserve experimental validation. Can the authors test their MD-based prediction by in vitro activity measurement with the Y69H mutant of RseP and a recombinant substrate where they could measure the kinetics of substrate cleavage for both variants of the enzyme (wt and mut)? This approach has been used by Akiyama et al. quite extensively and is feasible.

Minor issues

1. On page 4 '...that TNF α -(1-39) is a green colored peptide. ...' is an inacceptably inaccurate and misleading expression. The peptide is not green. Only its adduct with heme acquires a green colour. This must be corrected here and in any other place in the manuscript (such as on p. 6, top).

2. The manuscript is littered with incorrect English usage and typos, such as:
...their spectroscopically properties..., ...These finding are in...,... its proteolytical processing products...,
...generated by Leader peptidase..., ... This type of proteolytically activity is..., ...was not discernable
from..., ... adaptable for high-screening approaches aimed at...
Please use spell-checker and correct all mistakes.

Reviewers' comments and our point by point response

Reviewer #1 (Remarks to the Author):

The authors report a novel assay to study regulated intramembrane proteolysis (RIP). In prior work, the authors showed that RIP of TNF α changes its heme-binding characteristics. Here, they build on the previous work to create an innovative assay in which RIP changes the colony color of *E. coli* engineered to co-express an MBP-TNF α substrate (which is green due to high-affinity bis-thiolate ligation of heme) and the intramembrane metalloprotease RseP. Active RseP changes the colony color from light green to white (due to low-affinity penta-coordinated heme binding by the cleaved MBP-TNF α), whereas inactive RseP fails to change the colony color.

The breadth of applicability of the assay remains to be established. TNF α is normally cleaved by a signal peptide peptidase-like intramembrane aspartyl protease in eukaryotic cells. RseP cleaves the anti-sigma factor RseA, remnant signal peptides, and other substrates in *E. coli*. Therefore, the applicability of the assay may be broad, including (at least) the many intramembrane proteases that cleave Type 2 membrane proteins (with a single transmembrane segment and an extracytoplasmic C-terminus).

Importantly, the assay is amenable to high-throughput screening, which the authors demonstrate by screening rseP mutants generated by error-prone PCR. Screening of compound libraries for modulators of intramembrane proteases (that work in the assay) should be possible. High-throughput screening of compound libraries and intramembrane protease substitutions represent significant potential advances for the intramembrane protease field.

In general, the data are convincing and presented well in the manuscript. The colony-based assay is supported by SDS-PAGE of whole-cell extracts and by purification of MBP-TNF α substrate and product followed by spectroscopic and gel filtration/spectroscopic analyses. However, Figure 4 needs clarification and possibly additional experimental work, and the last section of the Results and Figure 6 add little to the manuscript and should be moved to the Supplemental Material.

I support publication after modifications to address the following comments.

We thank the reviewer for the positive evaluation of our manuscript and for the critical and helpful comments.

Major comments:

p. 6, l. 9 – do the authors have evidence that the shortened linker peptide increases the stability of the fusion protein? If so, the evidence should be

shown. If not, or if only the protein accumulation was measured, then the statement should be revised accordingly.

We have no experimental evidence that the linker peptide increases the stability of the fusion proteins. Following the reviewer's suggestion, we now write on p. 5 (last lane)

“... which should increase the proteolytic stability of the fusion protein.”
instead of “... to increase proteolytic stability of the fusion protein” .

p. 7, l. 6 – “product of I-CliP-mediated proteolysis in *E. coli*” is incorrect. Rather, the authors expressed MBPmut-TNFalpha-(1-34)-L31P, which was designed to mimic the predicted cleavage product.

We agree and now write on p. 6 (last lane) “mimicked product of I-CliP-mediated proteolysis in *E. coli*”.

p. 8, l. 5 from bottom to the end of the section – the presentation of Figure 4b-d is oversimplified and unclear. For 4b, “slightly smaller apparent molecular weight” is oversimplified since only the migration of the leading edge of the fusion protein bands differ and for the most part their migration overlaps. SDS-PAGE at higher resolution with less protein may resolve the cleavage product(s) from the uncleaved substrate in the RseP wt sample. The authors also ignore the presence of uncleaved substrate in the RseP wt sample in their presentation of 4c, yet the lower panel of 4d shows that dimer is present; however, it is unclear why the dimer elutes slightly later than the dimer in the RseP H22F sample (upper panel of 4d). The result seems to suggest that the RseP wt sample contains species with C-termini between residues 34 and 39, which form dimers. Also unclear is which fractions were subjected to UV/VIS spectroscopy in 4e. The figure legend states “fractions containing heme binding MBPmut-TNFalpha fusion proteins”, suggesting that the spectrum of the RseP wt sample is the fractions containing the dimer peak, but it seems more likely to be the monomer peak, given the shift to 392 nm. This needs to be clarified.

We agree with the reviewer that this section and presentation of Figure 4 have not been presented clearly enough. Therefore, we have made several changes in this section (p. 9-10) and in addition to Figure 4 and the Figure 4 legend (p. 28). The most important point is that heme binding species eluted from the gel filtration column are different for RseP H22F and RseP wt, not only because of the different elution volumes but also because of the different absorbance ratios A370/A450 for RseP H22F and RseP wt; Figure 4d). Only for **RseP H22F we did find heme binding dimers** (Figure 4e, upper panel). In the case of **RseP wt, monomeric heme binding processing products** are present (Figure 4e, lower panel). These may partially dimerize at higher protein concentrations (Fig. 4c; note that proteins are diluted during gel filtration). Our SDS gel data suggest that the MBP-TNFalpha fusion protein is slightly larger in the case of RseP H22F than in the case of RseP wt (compare with the contaminating protein migrating directly below the MBP fusions).

To answer this question unambiguously, we performed proteomics analyses of these proteins (see below).

Related to the preceding comment, a weakness of the manuscript is the incomplete characterization of the RseP cleavage product(s). Did the author attempt accurate mass determination of intact and/or trypsin-digested purified product(s)?

On p. 10-12 we have added the new paragraph “**RseP processing sites within substrate MBP_{mut}-TNF α -(1-39)-L31P**” to the revised version, which describes the mass spectrometric analysis (intact mass and MS/MS of tryptic peptides) of RseP processing products. By this mass spectrometric analysis (new Figure 5 and new Supplementary Figure 4), two major processing sites were found within the C-terminal nona-peptide of MBP_{mut}-TNF α -(1-39)-L31P (= remaining part of TNF α TMD). These new data confirm that RseP processing disrupts the dimerization motif of TNF α TMD and changes heme binding from bis-thiolate ligation to mono-thiolate ligation (leading to the observed color change).

The mass spectrometry proteomics data have been deposited to the ProteomeXchange Consortium via the PRIDE partner repository with the dataset identifier PXD038785 and 10.6019/PXD038785:

Username: reviewer_pxd038785@ebi.ac.uk **Password:** aUxIUQip

p. 10, l. 20 – the authors fail to comment on the results shown in Figure 6b. The Y69H and Y69H/Y482H variants of RseP are expected to produce products that migrate more slowly than those from wt and Y482H, but this is not convincing in Figure 6b. The band from Y69H/Y482H appears to migrate a bit more slowly, but less fusion protein is present, which may affect the appearance of the band. As noted above for Figure 4b, SDS-PAGE at higher resolution with less protein (and an equal amount of each protein) may resolve the fusion proteins in this case too. As noted above, accurate mass determination of intact and/or trypsin-digested purified product(s) may help.

In this case, we did not comment on the small size differences of the purified MBP fusions proteins in the manuscript, as different protein amounts were indeed applied to the gel. However, which proteins are processed by RseP and which are not can be reliably assessed from the UV/Vis spectra alone. Mass spectrometric analysis confirmed that changes in the UV/Vis spectra were due to truncation of C-terminal residues of MBP_{mut}-TNF α -(1-39)-L31P (see above).

The SDS gel of the purified proteins and UV/Vis spectra have been moved from the main text to Supplementary Figure 6.

Related to the preceding comment, the last section of the Results adds little to the manuscript. As mentioned in the Discussion, it was shown previously that RseP Y69P is severely impaired for proteolytic activity, so it is not surprising that Y69H does as well. The Y482H substitution in RseP has little effect. The AlphaFold prediction looks similar to the crystal structure depictions in preprint reference 20. The purpose of the MD simulations is not explained, no strong conclusions are made, and the significance of a lipid in the substrate binding groove is unclear. The last section of the Results and Figure 6 should be moved to the Supplemental Material.

To focus in the main manuscript more on the new iCliPSpy assay we moved five of the seven subfigures of Figure 6 to the supplement, but for several reasons we did not move the complete last section of the Results to the Supplemental Material:

- (1) By using the Y69H instead of the Y69P substitution we analyze the role of the side chain of Tyr69 and not the role of the whole β -loop for substrate binding [since the introduction of a Pro residue would disturb the higher-order structures of the MRE β -loop (Akiyama *et al.* 2015)].
- (2) The importance of the edge strand of RseP (residues Gly67 to Val70 including Tyr69 and part of the β -MRE) for substrate binding has already been described, and the crystal structure of RseP (published at the time of our studies) now shows that this edge strand can form an antiparallel beta sheet with the substrate TMD and that the bound inhibitor batimastat (substrate analogon) directly contacts Y69. It has also been shown that Y69 can be cross-linked to substrate molecules. Therefore, we believe that our experiments demonstrate the usefulness of the iCliPSpy assay (in combination with AlphaFold-based structures when structural data are not available, and MD simulations) for the identification of amino acid residues critical for enzyme activity and for the characterization of RseP and other intramembrane-cleaving proteases at the molecular level.
- (3) Tyr428 (also localized at the substrate binding groove) has not been characterized so far. It is not immediately obvious from the RseP structure that this residue plays no role in substrate binding.
- (4) We performed the MD simulations in a membrane environment, while for the crystal structure analysis of RseP the lipids were removed by solubilization with detergent. Although the structure of the hydrophobic core of RseP was then found to be very similar in both cases, the MD-simulated structure of membrane-embedded RseP helps to visualize the localization of Tyr69 at the membrane/cytosol interface.
- (5) We agree with the reviewer that the physiological importance of a lipid molecule placed next to/in the substrate binding groove by MD is not yet clear. However, this may be an important discovery as the lipid must be exchanged for the substrate protein during catalysis. Future work will show whether this bound lipid is “real” and whether lipid/substrate exchange is part of the RseP reaction mechanism.

p. 13, l. 12 – why do the authors assume that monomeric TNF α -(1-39) is bound by RseP and that the dimeric peptide is not bound? It seems conceivable that RseP binds to the dimer initially and then beta-strand addition dissociates the dimer, preventing high-affinity bis-thiolate ligation of heme and allowing cleavage of a monomer.

We now write (p. 17):

“It is not known what fraction of the substrate protein MBP_{mut}-TNF α -(1-39)-L31P forms a dimeric complex with heme and what proportion is monomeric, since we neither know the heme-binding constant nor the equilibrium constant for (heme-promoted) dimerization of TNF α -(1-39). Also, the dimerization mechanism itself is not elucidated, but we favor a model in which heme is first bound by monomeric TNF α -(1-39) and then this protein-heme complex binds a second TNF α -(1-39) monomer assisted by the SXXS dimerization motif. We assume (considering the sizes of the peptide binding groove and the active-site) that monomeric TNF α -(1-39) (without ligated heme) and not the dimeric (heme binding) one is finally cleaved by RseP. This does not exclude that initially present dimeric TNF α -(1-39) is bound by RseP and that beta-strand addition dissociates the dimer before cleavage. RseP-catalyzed cleavage of the dimerization motif within monomeric TNF α -(1-39) will result in decreased amounts of peptide TNF α -(1-39) available for (heme promoted) dimerization and increased amounts of cleaved peptides binding heme by only one Cys residue.”

p. 13, l. 19-23 – the authors should expand their discussion of the applicability of iCLiPSpy to other intramembrane proteases. For example, RseP has served as a model for a large family of PDZ domain-containing intramembrane metalloprotease found in nearly all bacteria, including many important pathogens where they affect expression of virulence genes.

We thank the reviewer for this suggestion. On p. 17-18 we have extended our discussion regarding the possible use of iCLiPSpy for the investigation of other intramembrane proteases including metalloproteases of pathogenic bacteria.

Minor points:

p. 2, l. 5 – change “belong to the most” to “are” since “most” is debatable

We changed the sentence as suggested.

p. 3, l. 2 – change “proteolytically” to “proteolytic”

Done.

p. 3, l. 10 – define “RIP” here or earlier

Done

p. 3, l. 3 from bottom – the GFG motif of RseP is also implicated in substrate binding, which should be mentioned and reference 33 should be cited

Done

p. 4, l. 1 – the original reference to RseP cleavage of signal peptides should also be cited (Saito et al. 2011 PNAS 108:13740)

Done

p. 4, l. 4 – “spectroscopically” should be “spectroscopic”

Done

p. 4, l. 8 – define “NTF”

Done

p. 6, l. 2 – change “putative” to “potential” since the authors hypothesize that TNFalpha-(1-39) may be a substrate, but no prior evidence supports this

Done

p. 6, l. 17 – change “leaking” to “leaky expression” (both occurrences)

Done

p. 6, l. 23 – suggest changing “uncolored” to “uncolored / white” here and using “white” thereafter to avoid confusion; change “scratched” to “scraped” here and elsewhere

Done

p. 7, l. 12 – change “recently published” to “published previously” since one of the papers cited was published in 2004

Done

p. 7, l. 16 - reference 33 should also be cited here

Done

p. 10, l.11 – “site” should be “side”

Done

p. 12, l. 4 – cite a reference at the end of the sentence

Done

p. 13, l. 2 from bottom – “high-screening” should be “high-throughput screening”

Done

p. 14, l. 3 – “mutations” should be “substitutions” here and elsewhere when describing amino acid changes

Done

p. 14, l. 7 – a single exchanged codon can alter the expression of a protein, so the sentence is incorrect. Are the authors indicating that genes were sequenced to ensure there were no other mutations?

- a) We now write: “To minimize differences in expression rates ...”
- b) Yes, on p. 19 l. 13 in the method section “Expression plasmids” it is stated that genes were sequenced: “The DNA sequence of all cloned genes was confirmed using appropriate primers.”

p. 14 – define TN buffer

Done

p. 15 – change “was now embedded” to “was then embedded”

Done

p. 22 – The y-axes of the graphs in Figure 4d should be explained (relative absorbance?)

Done

p. 30 – Figure 5a – arrows pointing to examples of one white colony and one green colony on each plate would be helpful since the difference is subtle

Done for the first plate, for the second plate we mention in the figure legend that the first row contains only white clones.

Reviewer #2 (Remarks to the Author):

The submitted paper “l-CliPSpy, a novel heme binding tag-based assay to characterize structure and function of the bacterial intramembrane protease RseP in vivo” describes a new assay to test the activity of intramembrane proteases in vivo in E.coli cells. Thomas Kupke and colleagues nicely show that a heme binding intramembrane substrate based on TNF α changes color upon cleavage by the target protease RseP. Protease and Substrate were co-expressed in E.coli cells. Overall, the study is conclusive, the experiments are described very well and were performed in a very reasonable and intuitive way. However, some minor issues need to be addressed before publication.

We thank the reviewer for the constructive and very helpful criticism.

1. One of the reporter constructs that the authors used carried an N-terminal His10-tag (construct 2). Importantly, this construct resulted in only weakly

colored cells (Fig. 2a), which is a very remarkable finding. However, the authors do not provide an explanation why they used this construct and why the cells were only weakly colored. An explanation would be very valuable. One could guess, based on Fig. 2c, color is reduced because of a reduced expression yield.

This is correct, as can be seen in Fig. 2, the color is reduced due to a lower expression yield. We have included this explanation in the amended manuscript.

2. In Figure 4d the authors show that on a SEC column the dimer peak of the reporter shifts to the right (lower molecular weight) upon cleavage by RseP wt. According to the authors hypothesis, the cleavage product should separate into monomers after cleavage, because of upon cleavage the dimerization motif is removed and thus heme promoted dimerization is prevented. I can understand that the monomer peak gets more prominent when RseP wt was used as compared to RseP H22F. How can this shift of the dimer be explained? And how does it fit to the assumption that RseP wt is cleaving monomeric TNF-reporter, as mentioned in the discussion on page 13?

We thank the reviewer for this comment and have made some changes in this section and in Figure 4 (and legend to Figure 4) to make the results more understandable to the reader (see p. 9-10). Please see our response to major comment 3 of reviewer 1.

3. In the discussion the authors write that they assume that RseP binds the monomeric and not the dimeric heme binding peptide. However, to me it is unclear how they come to this assumption. I think it would be good if the authors could explain this in more detail.

Please see our answer to major comment 7 of reviewer 1.

4. The authors did not mention whether it would be possible to do a quantitative analysis of the enzymatic activity using their assay. They suggest though that the assay may be used for screening of ligands, inhibitors or mutants of intramembrane proteases. However, for these screening applications it is important to have a quantitative read out. The “greenish” or white color of the E.coli colonies has to be measured by an instrument. How can such a quantitative readout be implemented? Maybe the reduction in the 450 nm peak could be used for quantification? It would help if the authors could show a quantitative measurement of the E.coli cell coloring, at least for one of their experiments.

Our results suggest that the color of the colonies can be used as quantitative measure in high-throughput screenings: green = inactive protease, white = active

protease and that the intensity of the color reflects the activity of the protease. The reviewer is correct that a high screening approach requires further development and depends on automated, digital imaging analysis to accurately monitor color changes of colonies. Establishing such a procedure is outside the scope of this work and would be done in collaboration with a laboratory specialized in setting up HTS procedures. Accordingly, we have only discussed this point as a possible future perspective.

5. The manuscript could benefit if the cleavage of the TNF-reporter would also be cross-confirmed in another experiment. Mass spectrometry could be a straight forward method to go for. Or, alternatively, adding a C-terminal Tag (e.g. strep-tag), which would allow to do western blot experiments. A C-terminal tag might however be incompatible with the cleavage by the target protease.

We agree with the reviewer and have added on p. 10-12 a new paragraph “**RseP processing sites within substrate MBP_{mut}-TNF α -(1-39)-L31P**”, describing the mass spectrometric analysis (intact mass and MS/MS of tryptic peptides) of the RseP processing products. Through this mass spectrometric analysis (new Figure 5 and new Supplementary Figure 4), two major processing sites were identified within the C-terminal nona-peptide of MBP_{mut}-TNF α -(1-39)-L31P (= remaining part of TNF α TMD). These new data confirm that RseP processing disrupts the dimerization motif of TNF α TMD, changes heme binding from bis-thiolate ligation to mono-thiolate ligation and leads to the observed color change.

The mass spectrometry proteomics data have been deposited to the ProteomeXchange Consortium via the PRIDE partner repository with the dataset identifier PXD038785 and 10.6019/PXD038785:

Username: reviewer_pxd038785@ebi.ac.uk **Password:** aUxIUQip

Minor issues:

6. The authors write that a crystal structure of RseP was determined. However, to interpret their results they use an AlphaFold generated model of RseP, rather than the crystal structure. Only in the discussion they mention that they do so because the crystal structure was not yet released by the wwPDB. This information is important and should be included in the results section, where the AlphaFold model is presented.

As suggested, we have added this information to the results section of the manuscript.

7. The authors used the detergent DDM to solubilize the membrane protein. However, this was only explained in the legend of Figure 2, not in the main text. I would recommend including the explanation for the DDM in the main text.

We have added this information to the first section of the results.

8. The authors suggest that their method can also be used as a screening assay for other intramembrane proteases. I'd recommend mentioning some specific examples of proteases that could be potential targets for this assay.

We thank the reviewer for this suggestion and have added interesting examples from pathogenic bacteria to the discussion section.

9. Overall, the manuscript is clearly written. It is easy to follow the content of the manuscript. However, the authors use numerous abbreviations. This is of course ok, but some of the abbreviations are not explained upon first use and some are not explained at all (e.g. RIP, TMD, ICD). Scientist working within this specific field might be familiar with these abbreviations. However, it will be hard for a non-specialist to understand the content of the paper if some important abbreviations are not explained.

We agree and have changed the text accordingly.

10. On page 14 it says "... TN buffer containing 2% of the detergent DDM (FROM)." I guess this is a typo.

This typo was corrected.

11. The authors provide sufficient details in the Material & Methods section and in the Supplement so the experiments can be certainly reproduced. The only thing I can criticize is that the composition of the TN buffer was not specified. At least I could not find it.

We have indeed forgotten to mention the composition of the buffer. This information has now been added.

12. Figure 5c: The orange color shows the location of active site residues. However, to me it is unclear where Tyr428 is exactly localized there. I'd recommend to e.g. label Tyr428 in another color or by an arrow.

We have added an arrow to show the localization of Tyr428.

13. Figure 6: The labeling of panels e, f and g is hard to read. I'd recommend increasing the contrast of the labeling.

We have changed some of the colors to increase the contrast.

Overall, the manuscript and the figures are clear and well-made and I believe that this article will be read with great interest by the scientific community.

Reviewer #3 (Remarks to the Author):

Kupke et al. present a novel in vivo assay for the activity of intramembrane metalloprotease RseP that they term I-ClIPSpy. The assay is based on an observation by the authors made earlier. The activity readout is indirect, and it is based on the transmembrane helix fragments of TNF produced by SPPL2a/b binds heme via thiolate or bis-thiolate ligation, where the two resulting types of complexes have different spectral properties. The reporter is designed such that it binds heme by the bis-thiolate ligation in the uncleaved state, while by the (single) thiolate mode in the RseP-cleaved state. Since the two states differ in their binding to heme and have different spectral properties (bis-thiolate adduct is green, while the thiolate one is colourless), heme binding and spectral (colour) changes report on RseP activity. The authors demonstrate the application of the assay by randomly mutagenizing RseP and isolating an inactive mutant with 11 mutations. Based on this analysis and investigation of the AlphaFold2 model of RseP they pinpoint the Y69 residue as participating in substrate binding. Finally, the authors propose that this assay can find general use for testing RseP inhibitors and activators, and for investigating other intramembrane proteases. This is a novel assay, but it should be characterized better, which would also help clarify how likely it is that it would find a more general use.

We thank the reviewer for the positive evaluation and for the constructive and very helpful comments.

I have the following criticisms and remarks to this study.

Major points

1. The membrane topology of the reporter is not entirely clear from the description in the paper. Since this is an engineered protein, it has to be properly described. Wild type TNF is a type II membrane protein, hence its C-terminus faces extracellular space. If signal peptide precedes MBP, TNF TMD would end up as a type I TMD, with its cleavage site region by RseP located at the intracellular side of the membrane. The MBP-based reporter could however also adopt a type II topology, dictated by the TNF transmembrane domain. Unfortunately, the exact design of the reporter is impossible to infer from the current description in the manuscript. Please add clearly the information on whether MBP contains its signal peptide, and specify the sequence and TMD boundaries of the TNF fragment. Clearly indicate this in Fig.1 and in the text/methods.

We apologize for the lack of detailed description of the design of the constructs. We have added the information that MBP is not preceded by a signal peptide (to ensure the correct topology of the reporter protein) in both the text and Figure 1. It is important to emphasize that our reporter protein MBP_{mut}-TNF α -(1-39)-L31P is not an

integral membrane protein, as only the nine N-terminal residues of the TNF α -TMD are present. The boundary between this partial TMD and the rest of TNF α is now shown in Figure 1. The complete amino acid sequence of the reporter protein MBP_{mut}-TNF α -(1-39)-L31P can be deduced from the DNA sequence provided in the supplement. Here we have listed the DNA sequence of all cloned genes to ensure reproducibility of our experiments.

2. Following on from point 1: is the reporter indeed membrane-localised? The efficiency of membrane targeting should be analysed by cellular fractionation (cytosol/membrane) and tightness of membrane association should be analysed (peripheral vs transmembrane) by sensitivity to high salt washes and detergent extraction. The topology of the reporter can be analysed experimentally by protease protection assay, which should probably be undertaken to characterize this novel reporter rigorously.

As mentioned in our response to point 1, the reporter protein is not an integral membrane protein. In the absence of a leader peptide, the MBP portion will be located in the cytosol (and not the periplasm) and the C-terminal nona-peptide originating from TNF α -TMD can associate with the membrane but cannot cross the membrane. We apologize for the misunderstanding as the required information was missing. The exact design of the reporter is now given in the amended manuscript. In addition, and complementary to the previous experiment, we also analyzed the cytosolic fraction, in which we could detect both the reporter protein and the processing products. We have added these experiments as a new Supplementary Figure 3. We assume that MBP_{mut}-TNF α -(1-39)-L31P can associate with the membrane due to the terminal nine TMD residues, but is also present in the cytosol due to the highly soluble MBP protein, however, cleavage by RseP only occurs when MBP_{mut}-TNF α -(1-39)-L31P is associated with the membrane and accessible to the membrane-embedded protease.

3. Cleavage site by RseP should be located within the TMD of TNF. How will heme bind to this intramembrane segment of TNF?

This is a very interesting question raised by the reviewer. Neither for TNF α -(1-39) nor TNF α -(1-34) a structure with bound heme is available. However, we know from spectroscopic data that heme is hexa-coordinated [also shown by electron paramagnetic resonance spectroscopy (EPR)] in the first case and is penta-coordinated in the second case (Kupke et al. 2020). The absorbance maximum at 370 nm (390 nm in presence of DDM) clearly shows that TNF α -(1-34) coordinates heme via the thiolate of residue Cys30 (Zhang and Guarente, EMBO J., 1995). The importance of Cys30 for heme binding has also been demonstrated by site-directed mutagenesis (Kupke et al., 2020). This cysteine residue is directly preceding the TMD residues. TNF α -(1-39) can dimerize in a heme-promoted way and in the dimer, Cys30 residues of both monomers ligate heme. The Ser34-X-X-Ser37 motif within the TMD residues is important for this dimerization, as shown in (Kupke et al., 2020) and is disrupted by RseP (cf newly included mass spectrometric data). Hydrophobic

side chains of the TMD residues may additionally contact the hydrophobic heme ring structure.

4. It seems that the assay employs endogenous heme from *E. coli* whose synthesis is stimulated by the δ -aminolevulinic acid in the growth medium. It must be made clearer in the manuscript. There is no mention of the growth medium containing δ -aminolevulinic acid) within the Methods.

Yes, the assay uses endogenous heme, because the lab strain *E. coli* K12 (in contrast to pathogenic *E. coli* strains) cannot take up heme. However, *E. coli* K12 can import the heme precursor δ -aminolevulinic acid from the medium. Therefore, addition of δ -aminolevulinic acid to the growth medium can increase the heme content of heme binding proteins (as also shown in Kupke et al., 2020). However, *E. coli* can also synthesize δ -aminolevulinic *de novo* and is not dependent on uptake of δ -aminolevulinic acid. The first version of our manuscript already indicated in the methods section for which of our experiments the growth medium was supplemented with δ -aminolevulinic acid. We now clarify in the methods section that no δ -aminolevulinic acid was added for the *in vivo* detection of RseP activity.

5. The cleavage site(s) in the TNF part of the reporter by RseP are only assumed based on the spectral properties of the heme conjugates and similarity to the SPPL2a/b ones. Cleavage site position(s) by RseP in the membrane localized reporter must be demonstrated by mass spectrometry.

We agree with the reviewer and have added on p. 10-12 a new paragraph “**RseP processing sites within substrate MBP_{mut}-TNF α -(1-39)-L31P**”, describing the mass spectrometric analysis (intact mass and MS/MS of tryptic peptides) of the RseP processing products. Through this mass spectrometric analysis (new Figure 5 and new Supplementary Figure 4), two major processing sites were identified within the C-terminal nona-peptide of MBP_{mut}-TNF α -(1-39)-L31P (= remaining part of TNF α TMD). These new data confirm that RseP processing disrupts the dimerization motif of TNF α TMD, changes heme binding from bis-thiolate ligation to mono-thiolate ligation and leads to the observed color change.

The mass spectrometry proteomics data have been deposited to the ProteomeXchange Consortium via the PRIDE partner repository with the dataset identifier PXD038785 and 10.6019/PXD038785:

Username: reviewer_pxd038785@ebi.ac.uk **Password:** aUxlUQip

6. The expression levels of the wild type and mutant RseP variants (e.g. in Fig. 3, Fig. 6) must be shown by immunoblot.

The expression of RseP was previously only detected by its proteolytic activity. The detection of this proteolytic activity is now also confirmed by mass spectrometry. In the revised version of our manuscript, we repeated the experiments shown in Figure 3 with Myc-tagged RseP wt and RseP H22F and subsequently confirmed the

expression of *rseP* by immunoblotting with a monoclonal anti-Myc antibody (to detect specifically over-expressed RseP and not the chromosomally encoded one). The mutant RseP G1 originated from error-prone mutagenesis performed with untagged *rseP* gene and is therefore also untagged. We did not tag the mutants Y69H and Y428H. In both cases, a single TAT codon in *rseP-wt* was exchanged for CAT, and the protein structures predicted by AlphaFold are almost identical to the wt structure. The expression of both genes in this case is confirmed by the *in vivo* activity (Figure 6d and 6f) and is already detectable without IPTG induction (as also shown for RseP wt and RseP-Myc wt).

7. How do the chemical identity, and TNF-peptide binding and spectral properties of the human heme (presumably used in the ref. 22 often referred to) and the E.coli heme (presumably used here) relate?

Heme (= heme B) is a ubiquitous cofactor with the same structure in prokaryotes and eukaryotes. (However, there are additional heme variants such as a, c and d and special heme variants in some bacteria, all these with special functions.) We have shown that synthetic TNF α -(1-39) peptide can be reconstituted with heme (= heme B) and forms bis-thiolate ligated heme complexes with the same absorbance properties as shown for the protein purified from *E. coli* (Kupke *et al.*, 2020).

8. Depending on the above details, how can this assay be generalized to other intramembrane proteases? It is based on using TNF and it is not clear if and which intramembrane proteases are likely to cleave TNF and in the expected topology. Can the heme binding peptide be engineered into other intramembrane protease substrates such that the heme binding readout can be used?

It should be possible to test other bacterial intramembrane cleaving proteases with the new iCliPSpy assay, especially site-2 metalloproteases. We have added some interesting examples from pathogenic bacteria to the discussion section.

We have also described CD74-NTF (N-terminal fragment of the type II transmembrane protein) as protein that binds heme via bis-thiolate ligation and that changes its heme binding mode by cleavage within the transmembrane domain (Kupke *et al.*, 2020). In principle, CD74-NTF (with a different TMD sequence than TNF α) can also be used to study intramembrane cleaving proteases (with different substrate specificities).

The specificity of intramembrane proteases seems to be low, so one substrate may be used for many different intramembrane cleaving proteases. On p. 17-18, we have added these points to the discussion section.

We assume that we need the motif - SRRC³⁰PFLSLFSFL³⁹ – for bis-thiolate heme ligation of TNF α -(1-39). Whether this motif can be fused directly to other proteins while retaining its heme ligation properties is currently unknown. So far, we have not investigated in detail which residues within this motif can be exchanged (without

disrupting heme bis-thiolate ligation) to test intramembrane cleaving proteases with different substrate specificities. However, we know that both serine residues are important (Kupke *et al.*, 2020).

9. How do the authors envisage that screening for inhibitors and activators can be accomplished with sufficient sensitivity and quantifiability in their current assay format? To what extent is the colony colour proportional to RseP activity?

Please see our response to comment 4 of reviewer 2.

10. The speculations on the role of Tyr69 in substrate interaction would deserve experimental validation. Can the authors test their MD-based prediction by in vitro activity measurement with the Y69H mutant of RseP and a recombinant substrate where they could measure the kinetics of substrate cleavage for both variants of the enzyme (wt and mut)? This approach has been used by Akiyama et al. quite extensively and is feasible.

We agree with the reviewer that these are very interesting questions, but feel that the proposed experiments are beyond the scope of our work, which was to introduce a new assay for the study of intramembrane cleaving proteases. In our manuscript we emphasize that our data confirm the results of the thorough enzyme study performed by Akiyama et al. and other published data, which allowed us to benchmark our assay. We hope to have convinced the reviewer that the iCliPSpy assay (in combination with mutagenesis and AlphaFold based structures and MD simulations) adds to the repertoire of available approaches for characterizing RseP and other intramembrane cleaving proteases at a molecular level.

Minor issues

1. On page 4 '...that TNF α -(1-39) is a green colored peptide. ...' is an inacceptably inaccurate and misleading expression. The peptide is not green. Only its adduct with heme acquires a green colour. This must be corrected here and in any other place in the manuscript (such as on p. 6, top).

We corrected this inaccurate and misleading expression in the manuscript.

2. The manuscript is littered with incorrect English usage and typos, such as: ...their spectroscopically properties..., ...These finding are in...,... its proteolytical processing products..., ...generated by Leader peptidase..., ... This type of proteolytically activity is..., ...was not discernable from..., ... adaptable for high-screening approaches aimed at... Please use spell-checker and correct all mistakes.

We apologize and have corrected the mistakes.

Reviewers' comments:

Reviewer #1 (Remarks to the Author):

In general, the authors did an excellent job of addressing the reviewers' comments. In particular, the presentation of Fig. 4 has been clarified and the characterization of cleavage products by two mass spectrometry approaches presented in the new Fig. 5 and Supplementary Fig. 4 strongly supports the author's interpretation that RseP cleaves the TNFalpha substrate in the partial transmembrane segment and leads to a change in heme binding that alters the E. coli colony color. The presentation of Fig. 6 (formerly Fig. 5) is appropriately shortened by moving panels to Supplementary Fig. 5. The writing is improved in terms of clarity, English usage, and explaining the importance of the work to a broad audience. However, the authors overstate what can be concluded from mass spectrometry of cleavage products generated in E. coli, because proteases other than RseP may contribute to the variety of products after RseP makes an initial cleavage. I support publication, but the mass spectrometry results and conclusions must be presented more carefully.

Comments:

p. 12, l. 9 from bottom – "It is likely that minor cleavage of the substrate protein by chromosomally encoded RseP (and other E. coli proteases) occurs" does not explain the absence of a small amount of the major processing product for RseP-Myc H22F in Table S2, which should be noted.

p. 12, last sentence – needs to be revised, because a processive cleavage mechanism is not proven by the data. Rather, E. coli proteases other than RseP may contribute to the variety of TNFalpha C-termini. Neither is it clear why the major processing products differ for RseP wt versus RseP-Myc wt (p. 12, l. 3-7). Perhaps the Myc tag or a difference in expression level?

p. 15, from "In vitro, we confirmed..." to the end of the paragraph – this part needs to be written more carefully. "In vitro" is confusing here because the authors are referring to "In vitro validation..." (subtitle on p. 8) of in vivo cleavage products. The authors did not purify RseP and TNFalpha, and perform in vitro reactions. Therefore, they cannot conclude that "Mass spectrometric analysis showed that RseP catalyzed trimming of MBPmut-TNFalpha-(1-39)-L31P at L/F-S peptide bonds releases the tripeptides S37FL39 and S34LF36" because it is possible that RseP cleaved just one residue from the C-terminus and other E. coli proteases generated a variety of products. It would be fair to say that RseP may engage in tripeptide trimming and point out the possible analogy to gamma-secretase, but in the last sentence of the paragraph, "determine trimming mechanism" should be "determine the mechanism".

Minor points:

p. 10, l. 1 – delete "used"

p. 11, l. 9 from bottom – change "is at all detectable, although there are no TNFalpha TMD residues left," to "is detectable, despite the absence of TNFalpha TMD residues,"

p. 11, l. 3 from bottom – change "(for methodic details see supplement)" to "(see Extended methods)"

p. 12, l. 5 from bottom – delete "used"

p. 14, l. 12 – should be "In the case"

p. 18, l. 7 – "(equals RseP)" should be "(a homolog of RseP)"

p. 18, l. 8 – "TCP" should be "TcpP" in both cases

p. 30 – Fig. 6 legend – l. 5 should be “them is shown”

p. 31 – Fig. 6 legend – l. 5 from bottom should be “name of the”

p. S7 – Supplementary Fig. 5 legend – l. 4 should be “cluster (see Extended methods) are”

Reviewer #2 (Remarks to the Author):

In the revised version of the manuscript the authors addressed all issues that I raised. The additional mass spec experiments and explanations that the authors included significantly improved the quality of the manuscript. Overall, the current version of the manuscript is much clearer. Therefore, I recommend accepting the revised version of the manuscript for publication.

Reviewer #3 (Remarks to the Author):

The revised version of Kupke et al. addresses largely all my factual points successfully. There just these remaining typos to correct:

p. 4: “...Ideally, such a reporter would change its spectroscopic characteristics upon proteolytical processing and also contain a tag that would allow affinity purification and detection. In a recent study, we used an E. coli expression model to identify TNFa, its proteolytical processing products...” should read “...Ideally, such a reporter would change its spectroscopic characteristics upon proteolytic processing and also contain a tag that would allow affinity purification and detection. In a recent study, we used an E. coli expression model to identify TNFa, its proteolytic processing products...”

p. 15: “...the trimming mechanism has recently be studied in detail...” should probably read “...the trimming mechanism has recently been studied in detail...”

Reviewers' comments and our point by point response

Reviewer #1 (Remarks to the Author):

In general, the authors did an excellent job of addressing the reviewers' comments. In particular, the presentation of Fig. 4 has been clarified and the characterization of cleavage products by two mass spectrometry approaches presented in the new Fig. 5 and Supplementary Fig. 4 strongly supports the author's interpretation that RseP cleaves the TNFalpha substrate in the partial transmembrane segment and leads to a change in heme binding that alters the E. coli colony color. The presentation of Fig. 6 (formerly Fig. 5) is appropriately shortened by moving panels to Supplementary Fig. 5. The writing is improved in terms of clarity, English usage, and explaining the importance of the work to a broad audience. However, the authors overstate what can be concluded from mass spectrometry of cleavage products generated in E. coli, because proteases other than RseP may contribute to the variety of products after RseP makes an initial cleavage. I support publication, but the mass spectrometry results and conclusions must be presented more carefully.

We thank the reviewer for the positive evaluation of our manuscript and for the additional critical and helpful comments.

Comments:

p. 12, l. 9 from bottom – “It is likely that minor cleavage of the substrate protein by chromosomally encoded RseP (and other E. coli proteases) occurs” does not explain the absence of a small amount of the major processing product for RseP-Myc H22F in Table S2, which should be noted.

One possible explanation is that we have different situations in “empty vector control” and “RseP-Myc H22F”. The concentration of endogenous active RseP wt in the membrane might be reduced due to the higher concentration of overexpressed inactive RseP-Myc H22F. In addition, both enzymes compete for the substrate.

As suggested by the reviewer, we now mention in the main text that the major processing product in the case of RseP-Myc H22F was not detected and have changed on page 12 (line 322) the sentence

“Small amounts of the major processing product of RseP, MBP_{mut}- ... SRRCPFLSLF³⁶, were also identified for the “Empty vector control” by LC-MS/MS analysis (Supplementary Table 2).”

to

“Small amounts of the major processing product of RseP, MBP_{mut}- ... SRRCPFSLSLF³⁶, were also identified for “Empty vector control” but not for “RseP-Myc H22F” by LC-MS/MS analysis (Supplementary Table 2).”

p. 12, last sentence – needs to be revised, because a processive cleavage mechanism is not proven by the data. Rather, *E. coli* proteases other than RseP may contribute to the variety of TNF α C-termini. Neither is it clear why the major processing products differ for RseP wt versus RseP-Myc wt (p. 12, l. 3-7). Perhaps the Myc tag or a difference in expression level?

We agree with the reviewer that *E. coli* proteases other than RseP may contribute to the proteolytic cleavage of MBP_{mut}-TNF α -(1-39)-L31P and had mentioned on page 12 auf our revised version that other *E. coli* proteases may further degrade the two major processing products. The major processing products do not differ for RseP wt versus RseP-Myc wt.

The following sentence (page 12, lines 306ff)

“However, when MBP_{mut}-TNF α -(1-39)-L31P was co-expressed with either *rseP wt* or *rseP-Myc wt*, MBP_{mut}-TNF α -(1-39)-L31P (= MBP_{mut}- ... CPFLSLFSFL³⁹) was cleaved between residues F36 and S37 of TNF α -(1-39)-L31P and between L33 and S34, resulting in the major processing products MBP_{mut}- ... CPFLSLF³⁶ and MBP_{mut}- ... CPFL³³, respectively (Fig. 5e, and Supplementary Fig. 4b, d, e).”

was not correctly worded by us and was therefore misunderstood by the reviewer. We apologize for this and have corrected the sentence accordingly to:

“However, when MBP_{mut}-TNF α -(1-39)-L31P was co-expressed with *rseP wt* or *rseP-Myc wt*, MBP_{mut}-TNF α -(1-39)-L31P (= MBP_{mut}- ... CPFLSLFSFL³⁹) was cleaved between residues F36 and S37 as well as between L33 and S34 of TNF α -(1-39)-L31P in both cases, resulting in the major processing products MBP_{mut}- ... CPFLSLF³⁶ and MBP_{mut}- ... CPFL³³ for both wt proteases (Fig. 5e, and Supplementary Fig. 4b, d, e).”

For *rseP wt* but not for *rseP-Myc wt*, small amounts of the processing product MBP_{mut}- ... CPFLSLFS³⁷ (1.94%; Table S2) were also detected by MS/MS analysis.

In order to include the cleavage product MBP_{mut}- ... CPFLSLFS³⁷ also in the main text, we have changed on page 12 (lines 310ff) the sentence

“Upon co-expression with *rseP wt*, we found small amounts of two additional cleavage products, MBP_{mut}- ... CPF³² and MBP_{mut}- ... CPFLS³⁴ (Fig. 5e, and Supplementary Fig. 4b).”

to

“Upon co-expression with *rseP wt*, we found small amounts of three additional cleavage products, MBP_{mut}- ... CPF³², MBP_{mut}- ... CPFLS³⁴ (Fig. 5e, and Supplementary Fig. 4b) and MBP_{mut}- ... CPFLSLFS³⁷ (Supplementary Table S2).”

Furthermore, on page 12 (lines 331-332) we have changed the sentence:

“The mass spectrometric analysis also provides insights into the processive cleavage mechanism and the specificity of RseP.”

to

“The mass spectrometric analysis suggests a processive cleavage mechanism of RseP and also gives insights into the specificity of RseP.”

p. 15, from “In vitro, we confirmed...” to the end of the paragraph – this part needs to be written more carefully. “In vitro” is confusing here because the authors are referring to “In vitro validation...” (subtitle on p. 8) of in vivo cleavage products. The authors did not purify RseP and TNFalpha, and perform in vitro reactions.

To avoid any misunderstanding, we have changed on page 15 (lines 401ff) the sentence:

“*In vitro*, we confirmed by UV/Vis spectroscopy and mass spectrometry that TNF α -(1-39)-L31P is indeed cleaved by RseP.”

to

“After purification of the MBP-tagged reporter protein we confirmed by UV/Vis spectroscopy and mass spectrometry that TNF α -(1-39)-L31P is indeed cleaved by RseP.”

Therefore, they cannot conclude that “Mass spectrometric analysis showed that RseP catalyzed trimming of MBPmut-TNFalpha-(1-39)-L31P at L/F-S peptide bonds releases the tripeptides S37FL39 and S34LF36” because it is possible that RseP cleaved just one residue from the C-terminus and other E. coli proteases generated a variety of products. It would be fair to say that RseP may engage in tripeptide trimming and point out the possible analogy to gamma-secretase, but in the last sentence of the paragraph, “determine trimming mechanism” should be “determine the mechanism”.

We think that it is very unlikely that “RseP cleaved just one residue from the C-terminus and other E. coli proteases generated a variety of products” because for RseP wt as well as for RseP-Myc wt we did not detect any MBP_{mut}- ... CPFLSLFSF³⁸ processing product neither by intact mass analysis nor by MS/MS analysis after tryptic cleavage.

Nevertheless, we have changed the sentence on page 15 (lines 403ff)

“Mass spectrometric analysis showed that RseP catalyzed trimming of MBP_{mut}-TNF α -(1-39)-L31P at L/F-S peptide bonds releases the tripeptides S³⁷FL³⁹ and S³⁴LF³⁶, whereas the native substrate RseA is cleaved by RseP between Ala108 and Cys109¹³.”

to

“Mass spectrometric analysis showed that RseP engages in trimming of MBP_{mut}-TNF α -(1-39)-L31P at L/F-S peptide bonds and releases the tripeptides S³⁷FL³⁹ and S³⁴LF³⁶, whereas the native substrate RseA is cleaved by RseP between Ala108 and Cys109¹³.”

and on page 15 (lines 409ff) the sentence

“In the future, MD simulations may also help to determine trimming mechanism of RseP more precisely and provide better insight into the mechanism of I-CLiPs in general.”

to

“In the future, MD simulations may also help to determine the mechanism of RseP more precisely and provide better insight into the mechanism of I-CLiPs in general.”

Minor points:

p. 10, l. 1 – delete “used”

Done

p. 11, l. 9 from bottom – change “is at all detectable, although there are no TNF α TMD residues left,” to “is detectable, despite the absence of TNF α TMD residues,”

Done

p. 11, l. 3 from bottom – change “(for methodic details see supplement)” to “(see Extended methods)”

Done

p. 12, l. 5 from bottom – delete “used”

Done

p. 14, l. 12 – should be “In the case”

Done

p. 18, l. 7 – “(equals RseP)” should be “(a homolog of RseP)”

Done

p. 18, l. 8 – “TCP” should be “TcpP” in both cases

Done

p. 30 – Fig. 6 legend – l. 5 should be “them is shown”

Done

p. 31 – Fig. 6 legend – l. 5 from bottom should be “name of the”

Done

p. S7 – Supplementary Fig. 5 legend – l. 4 should be “cluster (see Extended methods) are”

Done

Reviewer #2 (Remarks to the Author):

In the revised version of the manuscript the authors addressed all issues that I raised. The additional mass spec experiments and explanations that the authors included significantly improved the quality of the manuscript. Overall, the current version of the manuscript is much clearer. Therefore, I recommend accepting the revised version of the manuscript for publication.

We thank the reviewer for the positive evaluation of our manuscript.

Reviewer #3 (Remarks to the Author):

The revised version of Kupke et al. addresses largely all my factual points successfully. There just these remaining typos to correct:

We thank the reviewer for the positive evaluation of our manuscript

p. 4: “...Ideally, such a reporter would change its spectroscopic characteristics upon proteolytical processing and also contain a tag that would allow affinity purification and detection. In a recent study, we used an E. coli expression model to identify TNFa, its proteolytical processing products...” should read “...Ideally, such a reporter would change its spectroscopic characteristics upon proteolytic processing and also contain a tag that would allow affinity purification and detection. In a recent study, we used an E. coli expression model to identify TNFa, its proteolytic processing products...”

Done

p. 15: “...the trimming mechanism has recently be studied in detail...” should probably read “...the trimming mechanism has recently been studied in detail...”

Done